# Iron and sulphur cycling in the cGENIE.muffin Earth system model (v0.9.21)

Sebastiaan J. van de Velde[1,*], Dominik Hülse[1], Christopher T. Reinhard[2], and Andy Ridgwell[1]

[1]Department of Earth and Planetary Sciences, University of California, Riverside, CA 92521, USA
[2]School of Earth and Atmospheric Sciences, Georgia Institute of Technology, Atlanta, GA 30332, USA
[*]Current address: Bgeosys, Geoscience, Environment & Society, Université Libre de Bruxelles, Brussels, Belgium;
Operational Directorate Natural Environment, Royal Belgian Institute of Natural Sciences, Brussels, Belgium

**Correspondence:** sebastiaan.van.de.velde@ulb.be

**Abstract.** The coupled biogeochemical cycles of iron and sulphur are central to the long-term biogeochemical evolution of Earth's oceans. For instance, before the development of a persistently oxygenated deep ocean, the ocean interior likely alternated between states buffered by reduced sulphur ('euxinic') vs. buffered by reduced iron ('ferruginous'), with important implications for the cycles and hence bioavailability of dissolved iron (and phosphate). Even after atmospheric oxygen concentrations rose to modern-like values, the ocean continued, episodically, to develop regions of euxinic or ferruginous conditions, such as associated with past key intervals of organic carbon deposition (e.g. during the Cretaceous) as well as extinction events (e.g. at the Permian/Triassic boundary). A better understanding of the cycling of iron and sulphur in an anoxic ocean, how geochemical patterns in the ocean relate to the available spatially heterogeneous geological observations, and quantification of the feedback strengths between nutrient cycling, biological productivity, and ocean redox, requires a spatially-resolved representation of ocean circulation together with an extended set of (bio)geochemical reactions.

Here, we extend the 'muffin' release of the intermediate-complexity Earth system model cGENIE, to now include an anoxic iron and sulphur cycle (expanding the existing oxic iron and sulphur cycles), enabling the model to simulate ferruginous and euxinic redox states as well as the precipitation of reduced iron and sulphur minerals (pyrite, siderite, greenalite) and attendant iron and sulphur isotope signatures, which we describe in full. Because tests against present-day (oxic) ocean iron cycling exercises only a small part of the new code, we use an idealized ocean configuration to explore model sensitivity across a selection of key parameters. We also present the spatial patterns of concentrations and $\delta^{56}Fe$ and $\delta^{34}S$ isotope signatures of both dissolved and solid-phase Fe and S species in an anoxic ocean as an example application. Our sensitivity analyses show how the first-order results of the model are relatively robust against the choice of kinetic parameter values within the Fe-S system, and that simulated concentrations and reaction rates are comparable to those observed in process analogues for ancient oceans (i.e., anoxic lakes). Future model developments will address sedimentary recycling and benthic iron fluxes back to the water column, together with the coupling of nutrient (in particular phosphate) cycling to the iron cycle.

# 1 Introduction

The biogeochemical cycles of iron and sulphur are tightly coupled in the marine environment and play fundamental roles in the evolution and functioning of the Earth System (Raiswell and Canfield, 2012). In their main oxidised states, both sulphur (in the form of sulphate; $SO_4^{2-}$) and iron (in the form of iron (oxyhydr)oxides; $FeOOH$) are important electron acceptors in the oxi-

dation of organic matter in anoxic environments such as marine sediments or oxygen-deficient water columns (e.g., Black Sea, stratified lakes) (Thamdrup, 2000; Crowe et al., 2008; Raiswell and Canfield, 2012). In metabolising organic matter, microbial reduction of $SO_4^{2-}$ and $FeOOH$ produce reduced sulphide ($H_2S$) and ferrous iron ($Fe^{2+}$), respectively. When present at the same location, $H_2S$ and $Fe^{2+}$ combine into iron monosulphides and eventually pyrite ($FeS_2$) (Rickard, 1997, 2006), the burial of which couples the short-term, surface, cycles of Fe and S with their long-term, geological, cycles (Berner, 1989). Depending

on the relative ocean inventory of $H_2S$ vs. $Fe^{2+}$, the precipitation of $FeS_2$ can lead to an anoxic water body becoming either iron-rich ('ferruginous') or sulphide-rich ('euxinic') (Canfield, 1998; Poulton and Canfield, 2011) – $H_2S$:$Fe^{2+}$ ratios greater than 2:1 (the $S$:$Fe$ ratio in $FeS_2$) promoting euxinic conditions, and lower ratios promoting ferruginous conditions (Poulton and Canfield, 2011). For most of Earth's history, the ocean interior is thought to have been predominantly anoxic (Fig. 1; Lyons et al., 2014), which implies that reduced forms of iron and sulphur would have dominated the marine redox landscape (Poulton

and Canfield, 2011; Raiswell and Canfield, 2012).

Whether an anoxic water body becomes ferruginous or euxinic can have significant impacts on the availability of nutrients, with ferruginous conditions potentially leading to phosphate limitation, and euxinic conditions potentially leading to depletion of key biological trace elements (Van Cappellen and Ingall, 1996; Bjerrum and Canfield, 2002; Reinhard et al., 2013, 2017; Wallmann et al., 2019). Furthermore, before the advent of oxygenic photosynthesis, the productivity of marine ecosystems was

likely, at least partly, fuelled by the oxidation of $H_2S$ or $Fe^{2+}$ to their oxidised counterparts (Kharecha et al., 2005; Canfield et al., 2006; Ozaki et al., 2018; Thompson et al., 2019). As a result, the long-term evolution of the Earth system and the structure of marine ecosystems are closely tied to the evolution of the biogeochemical cycles of iron and sulphur. The ability to simulate in models the evolution of these cycles as well as geochemical distributions in the ocean hence becomes key to better understanding the early evolution of microbial ecosystems.

During the early stages of Earth's history (i.e., Archean to mid-Proterozoic), the ocean was rich in $Fe^{2+}$ (Fig. 1), which enabled the extensive deposition of banded iron formations (BIFs) and ferruginous shales during those Eons (Bekker et al., 2010; Planavsky et al., 2011; Sperling et al., 2015; Konhauser et al., 2017). BIFs are rare in sediments deposited after 1.8 billion years ago, which was initially hypothesised to reflect a transition to oxygen-rich bottom waters that prevented the build-up of soluble iron by removing it as insoluble iron oxides (Cloud, 1972; Holland, 1984). In a seminal paper, Canfield

(1998) suggested that the abundance of atmospheric oxygen during the Proterozoic (which was at most $\sim 10$ % of today; Canfield and Teske, 1996; Lyons et al., 2014) was too low to oxygenate the deeper waters of the ocean. Instead, he proposed that the disappearance of BIFs was driven by an increase of oceanic sulphate concentrations, allowing sulphide (produced by microbial sulphate reduction) to remove reduced iron from solution by the precipitation of $FeS_2$ (Canfield, 1998). Hence, this hypothesis implied that the deep ocean was euxinic for most of the Proterozoic. Since then, a large wealth of geochemical

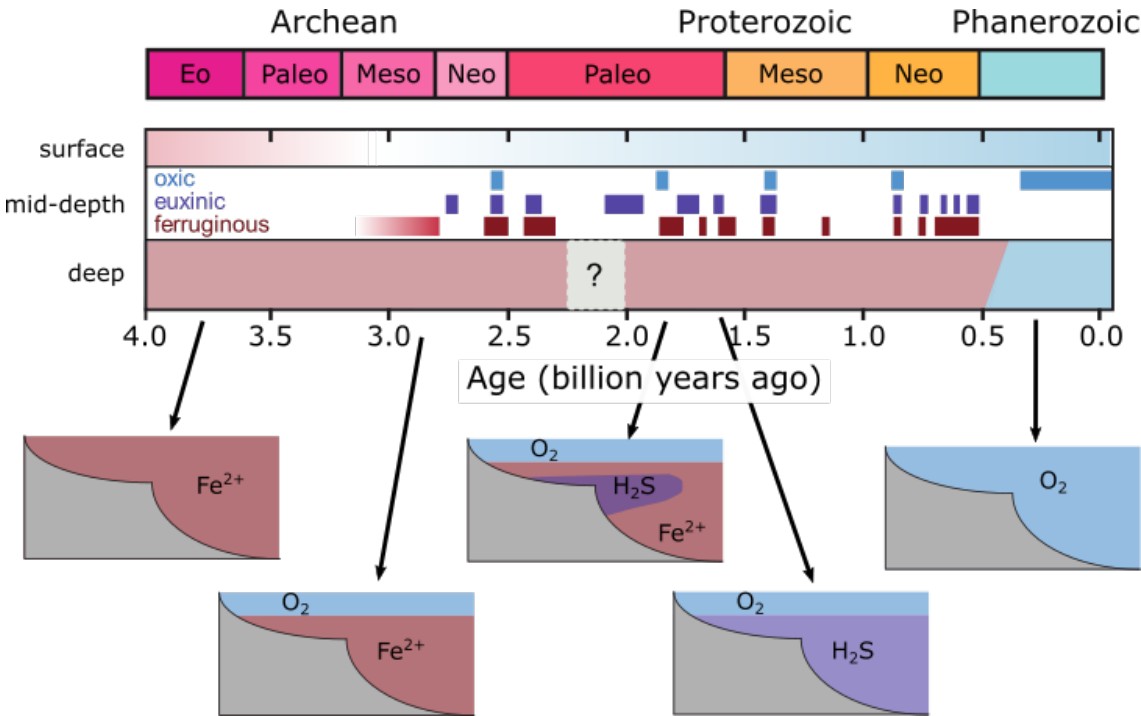

**Figure 1.** First-order evolution of Earth's ocean redox landscape. Insets show conceptual models of the spatial redox structure of the ocean at certain points in Earth's history. Based on (Poulton et al., 2010; Poulton and Canfield, 2011; Raiswell and Canfield, 2012). After (van de Velde et al., 2020b). See text for details.

proxy data has been collected, aided by the development of a sequential Fe extraction scheme that helps to differentiate between oxic, euxinic and ferruginous conditions by determining seven operationally defined iron mineral pools (Poulton and Canfield, 2005, 2011). These paleo-reconstructions of Archean and Proterozoic deposits helped to further refine the spatial and temporal history of ocean redox. In contrast to the earlier view, the Proterozoic Eon appears to have been dominated by ferruginous
5   conditions (Canfield et al., 2008; Planavsky et al., 2011; Poulton and Canfield, 2011; Guilbaud et al., 2015), likely interspersed with euxinic excursions along the shelf (Canfield et al., 2008; Poulton et al., 2010; Raiswell and Canfield, 2012) (Fig. 1). However, to date, these redox landscapes have been largely qualitative, since field-based observations are restricted by the limited number of available unaltered deposits of a certain point in time and sample acquisition for a depth transect is an enormously labour-intensive task (see e.g. Poulton et al., 2010). Because of these limitations, both the spatial redox pattern
10   and the exact nature of ferruginous versus euxinic conditions (i.e. the concentrations of $Fe^{2+}$ or $H_2S$) are still relatively unconstrained. The uncertainty with respect to the ocean redox conditions propagates into hypotheses on nutrient availability and ecosystem evolution (see, e.g.; Reinhard et al., 2017).

Whether the ocean interior is ferruginous or euxinic can significantly impact the supply of essential nutrients to surface ocean environments, and thus nutrient availability for primary producers (Van Cappellen and Ingall, 1996; Bjerrum and Canfield,

2002; Guilbaud et al., 2020). For example, iron oxides ($FeOOH$) can efficiently scavenge phosphate ($PO_4^{3-}$) from a water column, which has been suggested to limit oceanic phosphate availability in the past (Berner, 1973; Van Cappellen and Ingall, 1996; Bjerrum and Canfield, 2002; Jones et al., 2015). In contrast, a euxinic water column could potentially induce trace metal limitations by titrating out essential trace elements like iron or molybdenum (Reinhard et al., 2013; Wallmann et al., 2019). Indeed, lower nutrient availability - specifically phosphate - has been suggested to limit primary productivity for the majority of the Proterozoic (Bjerrum and Canfield, 2002; Reinhard et al., 2017; Ozaki et al., 2019). At the same time, nutrient availability is thought to have been critical in shaping and driving eukaryote evolution and proliferation (Brocks et al., 2017; Reinhard et al., 2020b). However, many of the most significant innovations in the history of Earth's biosphere have likely occurred in specific sites of the ocean that did not reflect the average state of the ocean as a whole (Nisbet and Sleep, 2001). Hence, the ability to reconstruct iron and sulphur cycling in a spatially explicit way is critical for exploring the relationships between biospheric evolution and changes in ocean redox.

None of the currently available suite of global models that explicitly represent biogeochemical cycling under low-oxygen marine environmental conditions include an extensive treatment of biogeochemical iron cycling (Ozaki et al., 2011; Laakso and Schrag, 2014; Hülse et al., 2017; Lenton et al., 2018; Reinhard et al., 2020a). Consequently, many essential interactions between the iron cycle and other elemental cycles (e.g. sulphur burial via pyrite precipitation) are abstracted and parameterised using techniques that may or may not be mechanistically robust across a range of scenarios. Moreover, most of the ocean models used to simulate the early stages of Earth's history are essentially box- or one-dimensional ocean models (Ozaki et al., 2011; Laakso and Schrag, 2014; Lenton et al., 2018) and are unable to resolve the spatial patterns necessary to start contrasting with the geological record (although some two- or three-box models attempt to resolve between the coastal zone and the open ocean; Laakso and Schrag, 2019; Thompson et al., 2019; Alcott et al., 2019). Indeed, previous simulations of past oceans with three-dimensional ocean models have already indicated the importance of spatial patterns in ocean redox (e.g. Olson et al., 2013), specifically when considering habitability for complex eukaryotic life (Reinhard et al., 2016, 2020b).

In this paper, we present the development of a coupled anoxic oceanic iron and sulphur cycle, embedded within the 'muffin' release of the carbon-centric Grid ENabled Integrated Earth system model, 'cGENIE' (note that the oxic cycle of both Fe and S already exists in previous versions; described in Tagliabue et al. (2016) and Ridgwell et al. (2007), respectively). The aim is to extend the functionality of cGENIE into regimes in which the ocean interior is pervasively anoxic, including much of Earth's Precambrian history and periods of significant perturbation to ocean redox during the Paleozoic and Mesozoic (during so-called 'Ocean Anoxic Events'; OAEs), where coupled iron-sulphur cycling dominated biogeochemical interactions in the ocean interior. Our extension explicitly accounts for the formation, burial, and isotopic compositions of key mineral phases used in paleoenvironmental reconstructions — specifically iron oxide ($FeOOH$), siderite ($FeCO_3$), greenalite ($Fe_3Si_2O_5(OH)_4$) and pyrite ($FeS_2$) (Poulton and Canfield, 2005, 2011) — which allows quantitative comparison with available data (e.g., Rouxel et al., 2005; Heard and Dauphas, 2020). In the next section (Section 2), we briefly discuss the cGENIE model framework, focusing on the features that are most relevant for our purpose of modeling pervasively anoxic oceans. Section 3 describes the included iron-sulphur reactions and the reasoning behind the chosen parameterisations. In Section 4 we discuss a modern con-

figurarion, an example configuration and a series of sensitivity experiments to the chosen parameterisation. Finally, in Section 5 we discuss model limitations and potential future developments.

## 2 The cGENIE.muffin Earth system model framework

cGEnIE is an earth system model of intermediate complexity (EMIC) which comprises a modular framework that incorporates
different components of the Earth system, including ocean circulation and biogeochemical cycling, ocean-atmosphere and ocean-sediment exchange, and the long-term (geological) cycle carbon and various solid-Earth derived tracers (Ridgwell et al., 2007; Ridgwell and Hargreaves, 2007; Colbourn et al., 2013; Adloff et al., 2020). Here, we use the current 'muffin' release that encompasses a range of developments and/or additions in the representation of: temperature-dependent metabolic processes in the ocean (Crichton et al., 2021), ocean-atmosphere cycling of methane (Reinhard et al., 2020a), marine ecosystems (Ward et
al., 2018), organic matter preservation and burial in marine sediments (Hülse et al., 2018), and geological cycles of weathering-relevant trace-metals and isotopes (Adloff et al., 2020).

    The climate component in cGENIE - C-GOLDSTEIN - consists of a 2-D energy-moisture balance model of the atmosphere coupled to a reduced physics (frictional geostrophic) 3-D ocean circulation plus dynamic-thermodynamic sea-ice model (see Edwards and Marsh, 2005; Marsh et al., 2011, for full descriptions). In addition to the simplified atmosphere, to further facilitate
the simulation of a relatively large number of interacting gaseous, dissolved, and solid tracers across atmosphere, ocean, and marine sediment (and land surface), cGENIE can be configured with a much reduced spatial and temporal resolution relative to most high-resolution ocean general circulation models (the default temporal resolution, which we use here, requires 48 timesteps per year in solving ocean circulation). While this precludes exploration of very detailed spatial patterns, it does provide a flexibility which is not available in more high-resolution models, and the relatively short run time of cGENIE
(around 1 day per 10,000 model years on a single CPU core) allows us to run many different model experiments and carry out comprehensive parameter sweeps and sensitivity analysis (and hence parameter tuning and model calibration). This aligns with our ultimate aim here which is to explore ocean biogeochemistry during periods of the Mesozoic (>65 million years ago), Paleozoic (>250 million years ago), and Precambrian (>540 million years ago). Most of these changes likely occurred on timescales exceeding 10,000 years, and at present we have virtually no information with respect to seasonality or detailed
spatial variability for many of these intervals (see, e.g., Poulton et al., 2010; Guilbaud et al., 2015). In addition, key boundary conditions and parameter values for these periods of Earth's history are often poorly constrained, necessitating large model ensembles in order to adequately assess the robustness of any given result.

### 2.1 Continental configuration and climatology

For the purpose of this study – implementing and characterising the coupled cycling of iron and sulphur in an anoxic ocean
– we adopt a deliberately idealised model configuration. We configure the ocean model on a 18x18 equal-area horizontal grid with 16 logarithmically spaced z-coordinate levels (Fig. 2). The horizontal grid is uniform in longitude (20° resolution) and uniform in the sine of latitude ($\sim$ 3.2° at the equator to 19.2° near the poles) (Fig. 2a). The layer thickness in the vertical grid

increases from 80.8 m at the surface to 765 m at the deepest layer (Fig. 2b). We adopt a 'Ridge World' set-up, with a thin strip of land connecting North and South poles (Fig. 2c), following Ferreira et al. (2010), which creates a single ocean basin with no circumpolar current (a little akin to the plate configuration prevailing during the late Permian). We apply idealised boundary conditions of zonally-averaged wind stress and speed, plus a zonally-averaged planetary albedo, all following Vervoort et al. (in review) (Fig. 3a-c). The solar constant is set to modern (1368.0 $W\,m^{-2}$). It is worth noting that our model setup is entirely abyssal, while in reality the continental slopes and shelves are important for the biogeochemical cycling of $Fe$ and $S$. We have chosen our idealized bathymetry and continental configuration as it generates a relatively simple ocean circulation and thus facilitates the interpretation of model output (see below). Our choice allows us to clearly illustrate the dependence of model output on the choice of parameters, whereas more elaborate continental configurations would introduce more spatial complexity and potentially obscure model sensitivity to parameter selection.

The physics parameters controlling the model climatology all follow the 16-level ocean based configuration assessed by Cao et al. (2009) (Table S1 in their manuscript), with the exception of the parameterisation controlling ocean mixing. When using the standard implementation of isoneutral diffusion and eddy-induced advection (Edwards and Marsh, 2005; Marsh et al., 2011), sharp vertical redox gradients simulated under extreme redox and high dissolved iron conditions resulted in unacceptably negative tracer concentrations at depth, particularly for dissolved iron. We hence disabled this parameterisation, reverting to the original un-adjusted horizontal+vertical diffusion physics configuration of Edwards and Shepherd (2002). We tested the modern cGENIE configuration of Cao et al. (2009) for both ocean mixing parameterisations, and found only a minimal difference in the large scale ocean circulation (i.e., a $\sim 2$ Sv stronger Atlantic meridional overturning circulation in the un-adjusted parameterisation).

## 2.2 The biological carbon pump

In this paper, we adapt a representation of biological export from the surface ocean driven by an implicit (i.e. unresolved) biological community with a highly parameterised uptake of nutrients in the photic zone. The description of the basic scheme can be found in Ridgwell et al. (2007), although we use the specific configuration (including iron co-limitation) following Tagliabue et al. (2016). The governing equations are summarised below.

Biological productivity in the euphotic zone (taken to be the surface layer in the ocean model) is controlled by dissolved phosphate ($PO_4^{3-}$) and dissolved iron ($Fe$, which we will define later) availability, the fractional ice coverage of each grid cell ($A$), mean ambient light and temperature. With the exception of the parameterisation of the temperature term, the equation for photosynthetic nutrient uptake is expressed in $M\,h^{-1}$ and follows Doney et al. (2006):

$$\Lambda = F_I \cdot F_T \cdot \min(F_{PO_4^{3-}}, F_{Fe^{3+}}) \cdot (1-A) \cdot \frac{\min([PO_4^{3-}], red_{P/Fe} \cdot [Fe])}{\tau_{bio}} \tag{1}$$

Rates of photosynthetic nutrient uptake are scaled to ambient dissolved nutrient concentrations ($PO_4^{3-}$ and $Fe^{3+}$), according to an optimal uptake timescale ($\tau_{bio}$=1521.6 hours; Meyer et al., 2016), and converting $[Fe]$ to the equivalent dissolved phosphate

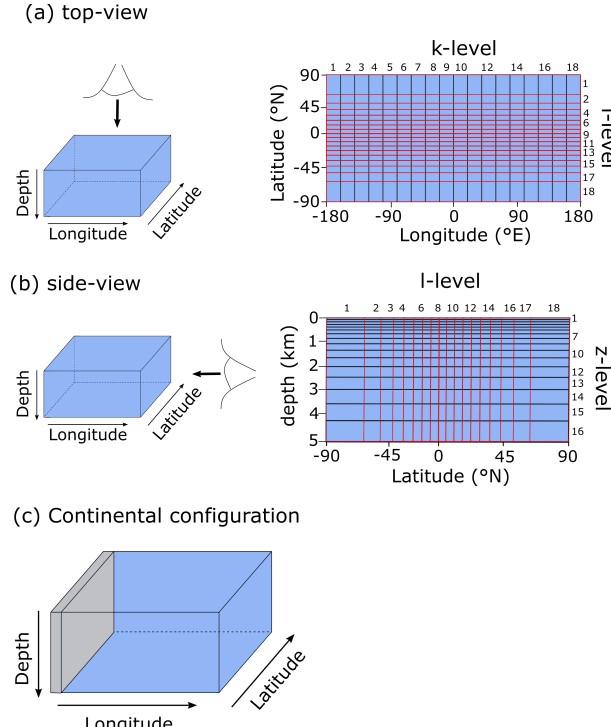

**Figure 2.** Schematic depiction of the cGENIE grid used for our simulations. The grid is 18x18x16, uniform in longitude and uniform in the sine of latitude. The layer thickness in the vertical grid increases from 80.8 m at the surface to 765 m at the deepest layer. (a) Top view of the cGENIE grid with indication of the layer numbers. (b) Side-view of the cGENIE grid with indication of the layer numbers. (c) Schematic picture of the continental configuration. 'Ridge World' has 1 continent which runs from pole to pole and extends from 0 to 20 °E (i.e., the width of a longitudinal grid-cell). The ocean is 5000 m deep everywhere.

concentration via the $Fe:P$ Redfield ratio ($red_{P/Fe}$). The various limitation terms (all unitless) are:

$$F_I = \frac{I}{I + \kappa_I} \tag{2}$$

$$F_{PO_4^{3-}} = \frac{[PO_4^{3-}]}{[PO_4^{3-}] + \kappa_{PO_4^{3-}}} \tag{3}$$

$$F_{Fe^{3+}} = \frac{[Fe]}{[Fe] + \kappa_{Fe}} \tag{4}$$

where shortwave irradiance $I$ is averaged over the entire mixed layer, and is assumed to decay exponentially from the sea surface with a length scale of 20 m, as per Doney et al. (2006). The $\kappa$ terms in each equation represent half-saturation constants for each limiting component ($\kappa_I$=40 $W\,m^{-2}$, $\kappa_{PO_4^{3-}}$=0.1 $\mu M$, $\kappa_{Fe^{3+}}$=0.1 $nM$) and are as used in Tagliabue et al. (2016). The

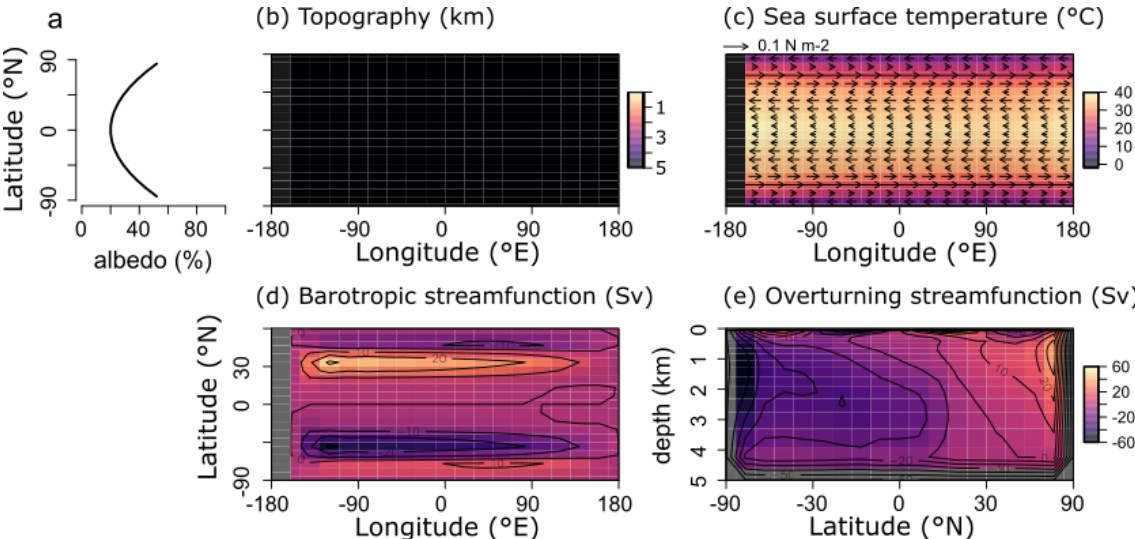

**Figure 3.** (a) Albedo, (b) Topography, (c) sea surface temperature and wind stress and (d,e) ocean circulation patterns for all model runs. (d) Barotropic streamfunction. (e) Overturning streamfunction. Positive values indicate clockwise circulation.

influence of temperature on biological export is parameterised as:

$$F_T = k_{T0} \cdot exp(\frac{T}{k_{eT}}) \tag{5}$$

where $k_{T0}$ (0.59) is a scaling constant, $k_{eT}$ (15.8°C) the *e*-folding temperature, and $T$ is the *in-situ* temperature (°C). The scaling constants give rise to an approximately factor two change per temperature change of 10°C (Reinhard et al., 2020a).

5    A proportion ($\nu = 0.66$) of $PO_4^{3-}$ taken up by biota is partitioned in dissolved organic phosphorus (DOP) while the remainder – as particulate organic phosphorus (POP) – is exported vertically out of the surface ocean. The value of $\nu$ has been assigned following the assumptions of the OCMIP-2 protocol (Najjar and Orr, 1999) (There is also an option for enacting temperature-dependent partitioning (and remineralisation) of DOP (Crichton et al., 2021), which presents an alternative to the fixed DOP/POP partitioning used here). Because the biological configuration used here does not resolve explicit standing

10   plankton biomass, the export flux of POP (in $mol\,m^{-2}\,h^{-1}$) is always equal to the rate of $PO_4^{3-}$ uptake:

$$F_{z=h_e}^{POP} = \int_{h_e}^{0} \rho(1-\nu)\Lambda dz \tag{6}$$

where $\rho$ is the density of seawater (in $kg\,m^{-3}$) and $h_e$ the thickness of the euphotic zone (=80.84 m). The particulate organic carbon (POC) export flux (in $mol\,m^{-2}\,h^{-1}$) is calculated using a fixed Redfield ratio as:

$$F_{z=h_e}^{POC} = 106F_{z=h_e}^{POP} \tag{7}$$

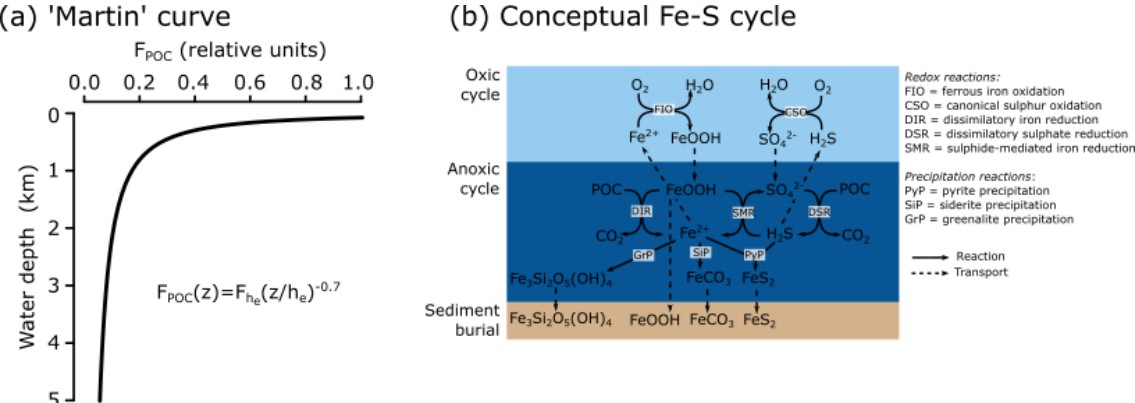

**Figure 4.** (a) Example of the 'Martin' curve for vertical organic matter fluxes in the ocean. (b) Conceptual description of the iron-sulphur cycle implemented in the model. For simplicity, only FeOOH (oxidised iron) and $Fe^{2+}$ (reduced iron) are depicted, but the model includes different iron species in both the oxic and anoxic states. Interaction between the methane cycle and sulphur cycle is omitted for clarity. See text for details.

After export from the surface grid cells, POC is remineralised instantaneously throughout the water column following a Martin -type curve (Martin et al., 1987), with a specified decay constant $b$ (=0.7) (Fig. 4a):

$$F_z^{POC} = F_{z=h_e}(z/h_e)^{-b} \tag{8}$$

The $b$-value of 0.7 is slightly higher than the global average of 0.6 estimated based on modern observations by Henson et al.
(2011), but leads to a better reconstruction of the distribution of $CaCO_3$ in deep-sea sediments (Wilson, pers. comm.).

## 3 Oceanic iron and sulphur cycling

The motivation behind this paper is to provide a tool that can aid understanding of the key interactions between the biological carbon pump, the oxygenation of the atmosphere and ocean, and the marine biogeochemical cycles of iron (Fe) and sulphur (S). In taking the first step towards this end, we construct a parsimonious model of ocean biogeochemical cycling based on
a simplified speciation scheme for both Fe and S, and only consider a relatively limited number of potential redox states. In this section we briefly describe the general conceptual cycle of Fe and S, as illustrated in Fig. 4b, and in the following subsections discuss the assumptions, reaction equations, and parameters for each cycle. The model equations that describe the biogeochemical reactions (summarised in Table 1) are given in Table 2 and the kinetic constants, their units and default values can be found in Table 3.
Redox cycling in the ocean (and sediments) is driven by the mineralisation of POC, which is produced in the photic zone by photosynthesis, and subsequently sinks through the water column (Ridgwell et al., 2007). A 'Martin'-type decay curve of organic matter flux with depth is prescribed in the model, such that regardless of the relative availability of different electron acceptors, the fraction of organic matter that will be degraded (relative to the amount of organic matter produced in the photic

**Table 1.** List of biogeochemical reactions included in reduced Fe-S scheme within the cGENIE model. A distinction is made between mineralisation ("Primary redox reactions"), re-oxidation of reduced products ("Secondary redox reactions") and precipitation reactions. The associated kinetic expressions are listed in Table 2. Note that we do not include here any details for reactions involving the methane cycle, though they are included in the simulations described here. For more information about the parametrisation and reactions of the methane cycle in cGENIE, we refer the reader to Reinhard et al. (2020a).

| Reaction number | Reaction name | Abbreviation | Equation |
|---|---|---|---|
| *Primary redox reactions* | | | |
| R1 | Aerobic respiration | AR | $CH_2O + O_2 \rightarrow CO_2 + H_2O$ |
| R2 | Dissimilatory iron reduction[1] | DIR | $CH_2O + 4FeOOH + 7H^+ \rightarrow HCO_3^- + 4Fe^{2+} + 6H_2O$ |
| R3 | Dissimilatory sulphate reduction | DSR | $CH_2O + \frac{1}{2}SO_4^{2-} + H^+ \rightarrow CO_2 + \frac{1}{2}\Sigma H_2S + H_2O$ |
| R4 | Methanogenesis | MG | $CH_2O \rightarrow \frac{1}{2}CO_2 + \frac{1}{2}CH_4$ |
| *Secondary redox reactions* | | | |
| R5 | Ferrous iron oxidation | FIO | $Fe^{2+} + \frac{1}{4}O_2 + H^+ \rightarrow Fe^{3+} + \frac{1}{2}H_2O$ |
| R6 | Canonical sulphide oxidation | CSO | $\Sigma H_2S + 2O_2 \rightarrow SO_4^{2-} + 2H^+$ |
| R7 | Sulphide-mediated iron reduction[1] | SMI$_{d/s}$ | $\Sigma H_2S + 8Fe^{3+}/FeOOH + 4H_2O \rightarrow SO_4^{2-} + 8Fe^{2+} + 10H^+$ |
| *Precipitation reactions* | | | |
| R8 | Iron oxide precipitation | IrP | $Fe^{3+} + 2H_2O \rightarrow FeOOH + 3H^+$ |
| R9 | Pyrite precipitation[2] | PyP | $FeS_p + \frac{3}{4}\Sigma H_2S + \frac{1}{4}SO_4^{2-} + \frac{1}{2}H^+ \rightarrow FeS_2 + H_2O$ |
| R10 | Siderite precipitation | SiP | $Fe^{2+} + CO_3^{2-} \rightarrow FeCO_3$ |
| R11 | Greenalite precipitation | GrP | $3Fe^{2+} + 2SiO_2(aq) + 5H_2O \leftrightarrow Fe_3Si_2O_5(OH)_4 + 6H^+$ |

[1] DIR can only occur with solid phase $FeOOH$, whereas SMI can occur with dissolved $Fe^{3+}$ and solid phase $FeOOH$. See text for more details. [2] The calculation of $FeS_p$ occurs implicitly from the concentrations of $Fe^{2+}$ and $\Sigma H_2S$ and the solubility of $FeS_{aq}$, see section 3.1.3 for more details.

zone) is depth-dependent. We avoid the alternative here – a fully kinetic set of equations where each electron acceptor is associated with a different rate of degradation – partly because of the additional set of poorly constrained (kinetic rate constant) parameters that would be required, and partly because to implement such a scheme effectively, requires knowledge about the composition of settling organic matter and how its relative reactivity changes with time (Ridgwell, 2011; LaRowe and Van Cappellen, 2011). Thus, the mineralisation rate is dependent on the magnitude of the $POC$ flux, which follows a 'Martin'-type decay (Fig. 4a), which determines the mineralisation rate ($R_{min}$) at each depth layer in the water column, and thus $R_{min}$ (in $M\ h^{-1}$) at depth z is calculated as,

$$R_{min}(z) = \frac{F_{POC}(z-1) - F_{POC}(z)}{\rho \Delta z} \tag{9}$$

where $\Delta z$ is the thickness of the grid cell at depth z (in $m$). Note that mineralisation of particulate organic matter only occurs below the surface layers; z > 1 in Fig. 2. The mineralisation of POC is coupled to the reduction of a terminal electron acceptor

(TEA). These TEAs are used according to decreasing energy yield (Froelich et al., 1979), and relative consumption rates are scaled with TEA concentration and the local abundance of inhibitory substances (i.e. a more energy-yielding TEA). Our mineralisation scheme includes aerobic respiration (AR, R1 in Table 1), dissimilatory iron reduction (DIR, R2 in Table 1), dissimilatory sulphate reduction (DSR, R3 in Table 1), and methanogenesis (MG, R4 in Table 1). Of these, AR and DSR existed in the original cGENIE biogeochemical framework (Ridgwell et al., 2007), while methanogenesis has been more recently added (Reinhard et al., 2020a). In the current paper, we ignore nitrate reduction (Monteiro et al., 2012).

The rate of TEA consumption is represented by a Michaelis-Menten type relationship with respect to TEA concentration that allows for a non-linear closure of the system:

$$f_{AR} = \frac{[O_2]}{K_{0,O_2} + [O_2]}$$

$$f_{DIR} = \frac{[FeOOH]}{K_{0,FeOOH} + [FeOOH]} \frac{K_{i,O_2}}{K_{i,O_2} + [O_2]}$$

$$f_{DSR} = \frac{[SO_4^{2-}]}{K_{0,SO_4^{2-}} + [SO_4^{2-}]} \frac{K_{i,FeOOH}}{K_{i,FeOOH} + [FeOOH]} \frac{K_{i,O_2}}{K_{i,O_2} + [O_2]}$$

$$f_{MG} = \frac{K_{i,SO_4^{2-}}}{K_{i,SO_4^{2-}} + [SO_4^{2-}]} \frac{K_{i,FeOOH}}{K_{i,FeOOH} + [FeOOH]} \frac{K_{i,O_2}}{K_{i,O_2} + [O_2]} \tag{10}$$

where $K_0$ are the half-saturation constants for the four primary redox reactions and $K_i$ are the inhibition constants that act on the less energetic redox reaction (Table 3). The consequence of this scheme is that in the oxic zone of the water column, aerobic respiration is responsible for nearly all of the POC mineralisation. When oxygen starts to become depleted, DIR, and subsequently DSR, become the dominant mineralisation pathways. When both iron oxides and sulphate are exhausted, MG represents the final mineralisation pathway. While the occurrence and parametrisation of DSR and MG in the water column are relatively straightforward, the possibility of DIR in the water column is more complex and is discussed in more detail in section 3.1.2.

As a consequence of organic matter remineralisation in the model, DIR produces ferrous iron ($Fe^{2+}$), which is re-oxidised when it comes into contact with oxygen (e.g., via upwelling of the reduced compound or downwelling of oxygen) via ferrous iron oxidation (FIO, R5 in Table 1) (Millero et al., 1987a). Similarly, any reduced sulphide (because we do not consider sulphide speciation explicitly, reduced sulphide is defined as $\Sigma H_2S = H_2S + HS^-$) that comes into contact with oxygen is re-oxidised via canonical sulphur oxidation (CSO, R6 in Table 1) (Millero et al., 1987b). Additionally, $H_2S$ is oxidised by reaction with oxidised iron via sulphide-mediated iron reduction (SMI, R7 in Table 1) (Fig. 4b; Canfield, 1992; Poulton et al., 2004; Mikucki et al., 2009). The oxidised form of iron, ferric iron ($Fe^{3+}$), will precipitate out as iron oxide ($FeOOH$) minerals (IrP, R8 in Table 1).

When $Fe^{2+}$ and $H_2S$ are simultaneously present in the water column they form dissolved $FeS$ ($FeS_{aq}$). Once $FeS_{aq}$ surpasses a solubility threshold of $\sim 2\,\mu M$ (Rickard, 2006), particulate $FeS_p$ can form, which further reacts with $H_2S$ to form the mineral pyrite ($FeS_2$) via pyrite precipitation (PyP, R9 in Table 1) (Rickard, 1997). Alternatively, when $Fe^{2+}$ accumulates past its saturation state with bicarbonate, siderite ($FeCO_3$) forms via siderite precipitation (SiP, R10 in Table 1) (Jimenez-Lopez et al., 2004; Jiang and Tosca, 2019). Finally, greenalite ($Fe_3Si_2O_5(OH)_4$) precipitates when $Fe^{2+}$ and dissolved silica

**Table 2.** List of kinetic rate expressions for the reactions included in the cGENIE model. All expressions are based on standard kinetic formulations in biogeochemical models. The values of the kinetic constants are listed in Table 3.

| Reaction number | Reaction name | Kinetic expression | Reference |
|---|---|---|---|
| R1 | Aerobic respiration | $f_{AR}R_{min}$ | [1],[2] |
| R2 | Dissimilatory iron reduction | $f_{DIR}R_{min}$ | [1],[2] |
| R3 | Dissimilatory sulphate reduction | $f_{DSR}R_{min}$ | [1],[2] |
| R4 | Methanogenesis | $f_{MG}R_{min}$ | [1],[2] |
| R5 | Ferrous iron oxidation | $k_{FIO}[\Sigma Fe^{2+}][O_2]$ | [2]-[4] |
| R6 | Canonical sulphide oxidation | $k_{CSO}[\Sigma H_2S][O_2]^2$ | [4]-[6] |
| R7 | Sulphide-mediated iron reduction[1] | $k_{SMI,d/s}[\Sigma H_2S]^{0.5}[Fe^{3+}]$ | [4],[7] |
| R8 | Iron oxide precipitation[2] | $k_{scav}[Fe^{3+}]F_{POC}(z)$ | [8],[9] |
| R9 | Pyrite precipitation | $k_{PyP}[FeS_p][\Sigma H_2S]$ | [10],[11] |
| R10 | Siderite precipitation | $k_{AFC}\,e^{(b_{AFC}\,log_{10}(IAP_{siderite}))}$ | [12] |
| R11 | Greenalite precipitation | $k_{greenalite}\,e^{(b_{greenalite}\,log_{10}(SI_{greenalite}))}$ | [13],[14] |

[1]Two different kinetic constants are used for the reactions between $Fe^{3+}$ and sulphide ($k_{SMI,d}$) and $FeOOH$ and sulphide ($k_{SMI,s}$), see text for details. [2] The formation of iron oxide minerals is parametrised according to Parekh et al. (2004), and is explained in more detail in section 3.1.1. References: [1] (Soetaert et al., 1996), [2] (van de Velde and Meysman, 2016), [3] (Millero et al., 1987a), [4] (Dale et al., 2015), [5] (Ridgwell et al., 2007), [6] (Zhang and Millero, 1993), [7] (Poulton et al., 2004), [8] (Parekh et al., 2004), [9] (Tagliabue et al., 2016), [10] (Rickard, 1997), [11] (Rickard, 2006), [12] (Jiang and Tosca, 2019), [13] (Tosca et al., 2015), [14] (Rasmussen et al., 2015)

($SiO_2$) are saturated with respect to greenalite precipitation (GrP, R11 in Table 1) (Tosca et al., 2015; Rasmussen et al., 2015). Pyrite, siderite and greenalite are subsequently buried in the sediment, together with any solid FeOOH that has not reacted (the half-life of iron oxides in euxinic waters is in the order of 10s - 100s of days, which is comparable to the residence time of a particle in the ocean; Poulton et al., 2004) (Fig. 4b). These four solid iron phases are the main burial phases for reactive iron, and form the basis of the Fe-speciation proxy used to reconstruct local redox conditions in past oceans (Poulton and Canfield, 2011). It should be noted however that some of these phases can undergo transformations to other phases in the sediment after deposition (such as greenalite to magnetite). In our current model set-up, no sedimentary processes are included, but future developments will address the sedimentary part of the $Fe$ and $S$ cycle.

Many of the iron and sulphur reactions can go to completion much faster than the biogeochemical time-step (1/24 yr in the example conceptual configuration) under typical modern or paleo geochemical conditions. This can create negative tracer concentrations because transport by ocean circulation acts concurrently on the tracer field in our numerical scheme. We hence place a limit on reactant consumption for any reaction in which a reactant would be completely consumed in a single time-step. Specifically, we prescribe a maximum time-scale for all geochemical reactions, for which in this paper we chose a value of 45 days.

## 3.1 The iron cycle

The cGENIE model already included the representation of a simplified iron cycle designed to account for iron limitation of biological productivity at the ocean surface (Tagliabue et al., 2016). However, because its initial use was to model the present-day oceanic iron cycle, it does not contain an anoxic iron cycle (as 95% of the today's ocean volume is oxic; Diaz and
Rosenberg, 2008), nor any coupling between the iron and sulphur cycles (e.g., via the precipitation of $FeS_2$). The absence of an anoxic iron cycle limits the suitability of cGENIE for simulating low-oxygen worlds in which the ocean interior is pervasively anoxic and iron and sulphur cycling would have dominated ocean biogeochemistry (Poulton and Canfield, 2011; Raiswell and Canfield, 2012). In addition to summarising the existing oxic cycle, we expand the model to include key processes operating in anoxic environments.

### 3.1.1 The oxic iron cycle

In oxygenated waters, iron is predominately present in its oxidised form ($Fe^{3+}$). The oxic iron cycle in cGENIE follows Parekh et al. (2004), and has been recently updated with a revised set of parameters, calibrated based on the present day iron cycle (Tagliabue et al., 2016). Briefly, in the previous scheme total dissolved $Fe^{3+}$ consists of free $Fe^{3+}_{free}$ and ligand-bound $Fe^{3+}_{ligand}$, with only the 'free' fraction being subject to scavenging on particulate organic carbon (POC) particles (whereas
both free and ligand-bound dissolved iron phases are assumed bioavailable and fuel primary productivity; see section 2.2). When scavenged on POC, $FeOOH$ sinks as a 'marine snow' particle through the water column. This scavenging mechanism is commonly used in reactive-transport models of ferruginous lakes or the modern ocean (see e.g. Taillefert and Gaillard, 2002; Tagliabue et al., 2016). Marine snow formation is what likely happens today, where relatively high production of organic matter drives the transport of chemically heterogeneous aggregates (containing iron oxides or barite; Dehairs et al., 1990; Balzano et
al., 2009). The scavenging rate of iron is a function of the concentration of $Fe^{3+}_{free}$ together with the magnitude of the POC flux from the grid cell, i.e.,

$$R_{IrP} = k_{scav}[Fe^{3+}_{free}]F_{POC}(z) \tag{11}$$

where $k_{scav}$ (=$1.43 \times 10^{-6} \ mol^{-1} \ m^2$) is a scavenging constant calibrated to the modern day distribution of $Fe$ (Tagliabue et al., 2016). We assume that the complexation reaction is always in equilibrium, which allows for the calculation of the amount
of $Fe^{3+}_{free}$ at each time step by the conservation equations

$$[Fe^{3+}] = [Fe^{3+}_{ligand}] + [Fe^{3+}_{free}]$$
$$[L_{total}] = [Fe^{3+}_{ligand}] + [L_{free}]$$
$$K^{FeL}_{sp} = \frac{[Fe^{3+}_{ligand}]}{[Fe^{3+}_{free}][L_{free}]} \tag{12}$$

where $K^{FeL}_{sp}$ is the stability constant of the $L - Fe^{3+}$ complex ($1.0 \times 10^{11} M^{-1}$; Table 3), $L_{free}$ is ligand unassociated with $Fe^{3+}$ and $L_{total}$ is the total amount of ligand.

Oxidised iron is highly insoluble in seawater and will rapidly form particulate oxides ($FeOOH$), driving down equilibrium dissolved $Fe^{3+}$ abundance to picomolar values without stabilisation by ligands (Liu and Millero, 2002). Thus, $Fe^{3+}_{free}$ will form colloidal or nanoparticulate iron oxides – $FeOOH$ – which can then adsorb on other particles or aggregate to bigger particles and sink through the water column (Raiswell and Canfield, 2012). In cGENIE, the pool of $Fe^{3+}_{free}$ can be thought of as a mix of purely dissolved $Fe^{3+}$ and a range of colloidal and nanoparticulate $Fe^{3+}$ phases that do not settle efficiently through the water column. The $Fe^{3+}_{free}$ pool can then further react or scavenge onto particles that allow it to more effectively settle through the water column.

In our new formulation of the Fe cycle in cGENIE, $Fe^{3+}_{free}$ (that is not stabilised by ligands) is still the phase that is scavenged by settling POC (using a fixed scavenging rate; Eq. 11), but explicitly assumed to be in the form of $FeOOH$ and can be used for DIR when oxygen becomes depleted (see Section 3.1.2). We also provide the option in the model for $Fe^{3+}_{free}$ to precipitate as a pure $FeOOH$ phase, without being associated with POC. In this case, dissimilatory iron reduction does not occur in the water column since both POC and $FeOOH$ are not associated within the same particle. This situation would be more appropriate in the case of high amounts of iron oxidation, and lower production of organic matter (which was potentially the case in the Archean; Thompson et al., 2019). This configuration is by default implemented using a numerical 'cut-off', where all $Fe^{3+}_{free}$ that passes the solubility threshold of 0.5 nM (Liu and Millero, 2002) is considered particulate and sinks through the water column.

To close the oceanic iron cycle, we have introduced a new dissolved iron species – $Fe^{2+}$ – any reduced $Fe^{2+}$ that is mixed into the oxic zone is being rapidly oxidise to $Fe^{3+}$ via ferrous iron oxidation (FIO),

$$Fe^{2+} + \frac{1}{4}O_2 + H^+ \rightarrow Fe^{3+} + \frac{1}{2}H_2O \tag{13}$$

where the rate equation can be expressed as

$$R_{FIO} = k_{FIO}[Fe^{2+}][O_2] \tag{14}$$

and where $k_{FIO}$ is a reaction rate constant ($k_{FIO} = 0.115 \times 10^6 M^{-1}h^{-1}$) (Table 2; Millero et al., 1987a). In the calculation of nutrient limitation of biological production at the surface, $Fe^{2+}$ is also assumed to be bioavailable, meaning that iron limitation is calculated from the total dissolved iron pool, equal to $Fe^{2+} + Fe^{3+}_{free} + Fe^{3+}_{ligand}$.

In summary, we have extended the basic representation of Parekh et al. (2004), that considered only a generic free 'dissolved' iron (that could be scavenged) and a ligand-bound iron phase, to now distinguish between the oxidation states of iron (and the tracers $Fe^{2+}$ and $Fe^{3+}$), with $Fe^{3+}$ being split (as previously) into a free (and scavengeable) and ligand-bound phase.

### 3.1.2 The anoxic iron cycle

In an anoxic water column, oxidised forms of iron can be reduced to $Fe^{2+}$ either via dissimilatory iron reduction or via sulphur-mediated iron reduction. We describe the two processes individually as follows.

**Dissimilatory iron reduction in the water column**

The majority of oxidised iron exists in particulate form due to the low solubility of $Fe^{3+}$, which poses a challenge for microorganisms performing dissimilatory iron reduction in the water column. Iron reducers need physical contact with the iron mineral to reduce iron (Gorby, 2006), which is difficult when both organic matter and $FeOOH$ are in particulate form and are dispersed separately throughout the aqueous medium. Even in a sediment column, where all particles are packed closely together, DIR generally requires sediment homogenisation by burrowing fauna to become volumetrically important (Thamdrup, 2000; van de Velde and Meysman, 2016). Indeed, in a sulphate-rich, ancient marine brine, $Fe^{3+}$ was found to be the terminal electron acceptor, but sulphur ultimately acted as a redox shuttle between organic matter and iron oxides (Mikucki et al., 2009). Studies of anoxic Lake Pavin suggest that most of the iron reduction occurs in the sediment, rather than the water column (Michard et al., 1994; Cosmidis et al., 2014), although the same studies indicated that manganese oxide reduction (which has a very similar mechanism and inhibition concentration as DIR; Lovley, 1991; Thamdrup, 2000) does occur in the water column. However, there is direct evidence for iron reduction occurring coupled to organic matter mineralisation in the water column in a range of other anoxic lakes, such as Lake Sammamish (Washington, USA; Balistrieri et al., 1992), Lake Cadagno (Switzerland; Berg et al., 2016), Lake Matano (Indonesia; Crowe et al., 2008), and Paul Lake (Michigan, USA; Taillefert and Gaillard, 2002). These studies suggested that DIR was coupled to either the oxidation of dissolved organic matter (Crowe et al., 2008), or reduction of iron in aggregates with organic matter. The latter has been experimentally shown to occur (Balzano et al., 2009). We consider this to be strong evidence that DIR can be coupled to POC oxidation in a water column, especially in the ocean where the residence time of particles is considerably longer than in a much shallower lake environment.

To model DIR in the water column, we use the mathematical expression of DIR in marine sediments, where $FeOOH$ is a common electron acceptor for organic matter mineralisation (Thamdrup, 2000). Dissimilatory iron reduction has an overall reaction stoichiometry of,

$$CH_2O + 4FeOOH + 7H^+ \rightarrow HCO_3^- + 4Fe^{2+} + 6H_2O \tag{15}$$

where it is implicitly assumed that every iron oxide particle consists solely of $Fe^{3+}$, rather than a mixture of redox states as is sometimes encountered at the interface of ferruginous lakes (Zegeye et al., 2012). This assumption greatly simplifies the reaction scheme and parameter set. Dissimilatory iron reduction generally becomes limited when concentrations of FeOOH drop below $\sim 30 \ 10^{-3} \ M$ (Van Cappellen and Wang, 1996; Thamdrup, 2000). This limitation is expressed as $\frac{[FeOOH]}{K_{0,FeOOH}+[FeOOH]}$ in Eq. 10 and follows the conventional limitation-inhibition scheme (Soetaert et al., 1996), where the parameter $K_{0,FeOOH}$ expresses the concentration at which DIR occurs at half of its maximum rate (the maximum rate is set by $R_{min}$; see Eq. 10 and Table 2). In marine sediments, DIR generally occurs before sulphate reduction since it is a more energy-yielding electron acceptor (Thamdrup, 2000), although the wide range of reactivities for different iron oxide minerals often leads to an overlap of DIR and DSR zones (Postma and Jakobsen, 1996). This limitation is expressed as $\frac{[K_{i,FeOOH}]}{K_{i,FeOOH}+[FeOOH]}$ in Eq. 10, where $K_{i,FeOOH}$ expresses the concentration above which FeOOH inhibits other mineralisation pathways (DSR and MG). In early diagenetic models, this concentration is generally assumed to be identical to the limitation parameter $K_{0,FeOOH}$ (Van Cappellen and Wang, 1996; Soetaert et al., 1996; van de Velde and Meysman, 2016). As a baseline value, we assume that $K_{0,FeOOH}$ and $K_{i,FeOOH}$ both equal $10^{-3} \ M$ (comparable to the inhibition concentration in marine sediments), but as ex-

plained above, these parameter values are subject to a high degree of uncertainty. Therefore, we include parameters $K_{0,FeOOH}$ and $K_{i,FeOOH}$ in our model sensitivity testing (section 4.3).

**Sulphur-mediated iron reduction**

Oxidised iron in the ocean can also be reduced via sulphur-mediated iron reduction (SMI), which follows the stoichiometry (Poulton et al., 2004):

$$\Sigma H_2S + 2Fe^{3+} \rightarrow S^0 + 2Fe^{2+} + 2H^+ \tag{16}$$

We are not explicitly modelling elemental sulphur ($S^0$), but assume that it becomes quantitatively disproportionated into $H_2S$ and $SO_4^{2-}$ (Finster et al., 1998),

$$S^0 + H_2O \rightarrow \Sigma \frac{3}{4}H_2S + \frac{1}{4}SO_4^{2-} + \frac{1}{2}H^+ \tag{17}$$

which then leads to the overall stoichiometry

$$\Sigma H_2S + 8Fe^{3+} + 4H_2O \rightarrow SO_4^{2-} + 8Fe^{2+} + 10H^+ \tag{18}$$

Note that $[Fe^{3+}]$ in Eq. 19 can represent dissolved $Fe_{total}^{3+}$ or solid $FeOOH$. The assumption of quantitative disproportion-ation implies that pyrite precipitation is not closely coupled to the reaction between $\Sigma H_2S$ and $Fe^{3+}$, but only occurs via precipitation of $FeS_p$ with $\Sigma H_2S$ (see Section 3.1.3). Laboratory experiments have shown that this assumption is valid for aquatic systems where $Fe^{3+}$ is not in excess with respect to $\Sigma H_2S$ (Wan et al., 2017), which is the case for most modern marine systems. We contend that this is also a valid assumption for water-column chemistry for most of Earth's history, as rapid settling of oxidised, particulate $FeOOH$ through the water column would prevent high concentrations of $Fe^{3+}$ (in the water column). To achieve an excess of $Fe^{3+}$ over $\Sigma H_2S$, the $\Sigma H_2S$ concentrations would have to be even lower, leading to very negligible rates of iron reduction. However, this reaction pathway would likely become important in the sediment of low-sulphate ocean (i.e. periods of Archean time).

The kinetic rate for reaction 18 can then be expressed as (Poulton et al., 2004):

$$R_{SMI} = k_{SMI,d/s}[\Sigma H_2S]^{0.5}[Fe^{3+}] \tag{19}$$

We assume that the dissolved form is highly reactive with sulphide (a reaction time of $\sim 5$ minutes, which is comparable to the reactivity of freshly precipitated hydrous ferric oxide) and has a reaction rate constant of $k_{SMI,d} = 2.64 \times 10^2\,M^{-0.5}\,h^{-1}$ (Poulton et al., 2004). The solid form of $FeOOH$ can represent a number of different oxidised iron minerals which are reactive towards sulphide on timescales ranging from hours to hundreds of days (Poulton et al., 2004). For our baseline simulations, we assume that all $FeOOH$ precipitates as lepidocrocite, which has a reactivity constant of $k_{SMI,s} = 1.98 \times 10^0\,M^{-0.5}\,h^{-1}$ (Poulton et al., 2004). Lepidocrocite is less crystalline than the other (non-hydrous) iron oxides and precipitates when the rate of $Fe^{3+}$ supply is low relative to the rate of precipitation, as is generally the case in natural systems (Crosby et al., 1983). We

discuss the model sensitivity to choices of $k_{SMI,s}$ in Section 4.3. The reduction of $Fe^{3+}$ produces $Fe^{2+}$, which can either be re-oxidised when it comes in contact with $O_2$ (see section 3.1.1) or can form reduced minerals.

Both reduction and oxidation reactions of dissolved iron, even with dissolved $O_2$ and $\Sigma H_2 S$ in nM concentrations, can proceed very rapidly and are hence subject to the geochemical reaction rate limitation described earlier. Furthermore, because these two reactions form a coupled oxidation/reduction pair, we limit the fastest reaction according to a 45 day time-scale, but limit the slower one in proportion to their relative unmodified reactions rates. We thereby simulate an equilibrium partitioning between $Fe^{2+}$ and $Fe^{3+}$ according to ambient dissolved oxygen and sulphide concentrations.

### 3.1.3 Reduced iron mineral formation

Reduced $Fe^{2+}$ can form complexes with a number of inorganic ligands that are common in seawater, including $Cl^-$, $SO_4^{2-}$ and bicarbonate (Millero et al., 1995). Under anoxic conditions, however, the most important inorganic ligand is free sulphide (Rickard, 2006), which is produced by sulphate reduction. Together, dissolved $Fe_{free}^{2+}$ and the aqueous iron-sulphide complex ($FeS_{aq}$) make up $\sim 100$ % of the total dissolved iron pool in sulphidic-anoxic seawater (Fig. 5a). The thermodynamic equilibrium can be calculated as

$$K_{sp}^{FeS_{aq}} = \frac{[Fe^{2+}][\Sigma H_2 S]}{[FeS_{aq}]} = 10^{-5.08} M \tag{20}$$

which was obtained from the visualMINTEQ database (Gustafsson, 2019), and is based on stability constants calculated by Luther et al. (1996).

Since dissolved $Fe_{free}^{2+}$ and the $FeS_{aq}$ complex are the dominant dissolved forms of reduced ferrous iron in natural waters (in anoxic seawater devoid of sulphide, dissolved $Fe_{free}^{2+}$ still represents $\sim 80$ % of the dissolved $Fe^{2+}$ pool; Fig. 5b), we choose to only consider those two species. We do not explicitly model the $FeS_{aq}$ complex, but calculate the thermodynamic equilibrium before each reaction proceeds, implicitly assuming that it is reached much faster than any of the kinetic reactions in our reaction set (as is commonly done for the complexation of $Fe^{3+}$; see section 3.1.1). This is important, since $FeS_{aq}$ forms particulate $FeS_p$, which is the precursor of pyrite ($FeS_2$) (Rickard, 1997, 2006). Furthermore, siderite or greenalite can only precipitate from $Fe_{free}^{2+}$ (Tucker, 2001; Tosca et al., 2015), and their precipitation rates are consequently dependent on the dissolved $Fe_{free}^{2+}$ concentration.

Before each precipitation reaction, we calculate the concentration of $FeS_{aq}$, $Fe_{free}^{2+}$ and $H_2 S_{free}$ using equation 20. We then compare the $FeS_{aq}$ concentration to the solubility threshold ($10^{-5.7}\ M$; Rickard, 2006). If $FeS_{aq}$ surpasses the solubility value, the fraction above the solubility concentration precipitates instantaneously as particulate iron monosulfide ($FeS_p$) (Suits and Wilkin, 1998). Note that here we consider $FeS_p$ solubility to be independent of pH, which should be valid for seawater pH above 7 (Rickard, 2006). Pyrite can then form locally via reaction between $FeS_p$ and free $\Sigma H_2 S$, with the production of free hydrogen gas (as $S^{2-}$ in $\Sigma H_2 S$ and $FeS_p$ has to be oxidised to $S^{1-}$ in $FeS_2$; Rickard, 1997),

$$FeS_p + \Sigma H_2 S \rightarrow FeS_2 + H_2 \tag{21}$$

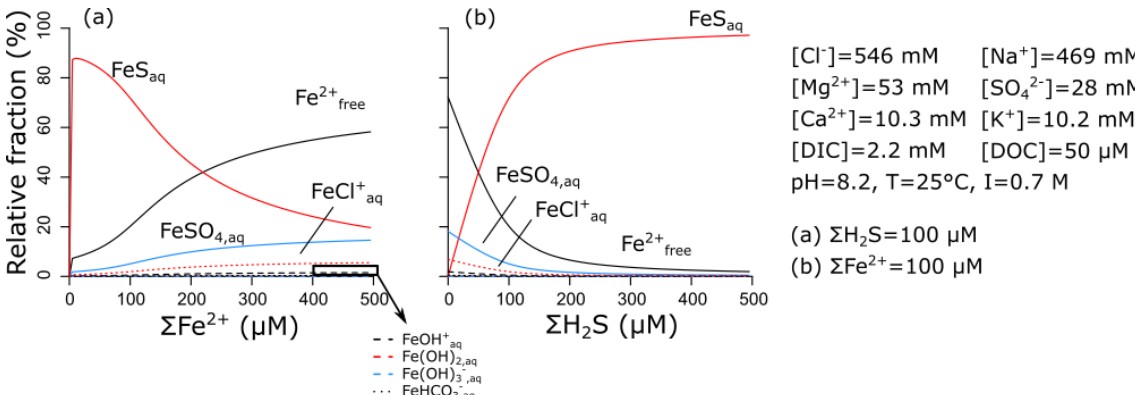

**Figure 5.** Relative importance of different dissolved Fe complexes, obtained by visualMINTEQ (Gustafsson, 2019). Concentrations of other constituents are chosen to respresent average seawater, with dissolved organic carbon (DOC) concentration from Sharp et al. (1995).

Since hydrogen-gas is highly reactive it is almost instantaneously oxidised, with the reduction of an electron acceptor, like $O_2$, $SO_4^{2-}$ or $CO_2$. Given that pyrite precipitates under sulphidic conditions, oxygen is absent, and sulphate is the most likely electron acceptor

$$H_2 + \frac{1}{4}SO_4^{2-} + \frac{1}{2}H^+ \rightarrow \frac{1}{4}\Sigma H_2 S + H_2 O \tag{22}$$

To preserve the redox balance while avoiding the need to model an additional state variable ($H_2$), we can combine reactions 21 and 22 to

$$FeS_p + \frac{3}{4}\Sigma H_2 S + \frac{1}{4}SO_4^{2-} + \frac{1}{2}H^+ \rightarrow FeS_2 + H_2 O \tag{23}$$

By using equation 23 we implicitly assume that the hydrogen gas produced during pyrite precipitation immediately reacts with $SO_4^{2-}$. Since pyrite formation is not an equilibrium reaction (pyrite minerals are stable in seawater), the reaction of pyrite is

described as a kinetic reaction with a second-order dependency on $[FeS_p]$ and $[\Sigma H_2 S]$, with $k_{PyP} = 0.3708 \times 10^0 \, M^{-1} \, h^{-1}$ (Rickard, 1997). The rate equation for $FeS_2$ precipitation can then be written as

$$R_{PyP} = k_{PyP}[FeS_p][\Sigma H_2 S] \tag{24}$$

Any $Fe^{2+}$ that is not complexed with sulphide ($Fe^{2+}_{free}$) can form siderite ($FeCO_3$), an iron-carbonate mineral (Tucker, 2001).

$$Fe^{2+}_{free} + CO_3^{2-} \rightarrow FeCO_3 \tag{25}$$

Recent experiments by Jiang and Tosca (2019) have shown that the precipitation of iron-carbonate phases is controlled by the formation of amorphous Fe carbonate (AFC), which has the stoichiometric formula $FeCO_3(OH)_{1/2}$. Since the thermo-dynamic data required to calculate the mineral solubility product of AFC is currently lacking, Jiang and Tosca (2019) have

determined the rate of AFC precipitation based on the ion activity product (IAP),

$$IAP_{AFC} = \gamma_{Fe^{2+}}[Fe^{2+}_{free}] \, \gamma_{CO_3^{2-}}[CO_3^{2-}] \, (\gamma_{OH^-}[OH^-])^{0.5} \tag{26}$$

where $\gamma$ represents the activity coefficient of a given ion in seawater (Table 3). The rate-limiting step in the precipitation reaction is the spontaneous nucleation of AFC, and hence the empirically derived rate of AFC precipitation follows an exponential function that describes the nucleation rate from water (Steefel and Van Cappellen, 1990; Jiang and Tosca, 2019),

$$R_{SiP} = k_{AFC} \, e^{b_{AFC} \, log_{10} IAP_{AFC}} \tag{27}$$

where $k_{AFC}$ is $1.963 \times 10^{-14} \, M \, h^{-1}$ and $b_{AFC}$ equals 9.042 (Table 3).

Another reduced iron mineral potentially important for anoxic, Fe-rich environments is the iron-silicate mineral greenalite ($Fe_3Si_2O_5(OH)_4$) (Tosca et al., 2015):

$$3Fe^{2+}_{free} + 2SiO_2(aq) + 5H_2O \rightarrow Fe_3Si_2O_5(OH)_4 + 6H^+ \tag{28}$$

The precipitation rate of greenalite is dependent on its degree of supersaturation (i.e. its saturation index), which can be calculated based on its IAP,

$$IAP_{greenalite} = \frac{(\gamma_{Fe^{2+}}[Fe^{2+}_{free}])^3 \, (\gamma_{SiO_2}[SiO_2])^2}{(\gamma_{H^+}[H^+])^6} \tag{29}$$

where $\gamma$ represents the activity coefficient of an ion in seawater (Table 3), and $K_{sp}^{greenalite}$ its solubility product (Rasmussen et al., 2015)

$$SI_{greenalite} = log_{10}\left(\frac{IAP}{K_{sp}^{greenalite}}\right) \tag{30}$$

where $K_{sp}^{greenalite}$ equals $3.98 \times 10^{27} \, M^{-1}$ (Tosca et al., 2015). The rate equation follows an exponential function that describes the nucleation rate from water (as in siderite, see above; Rasmussen et al., 2015; Jiang and Tosca, 2019)

$$R_{GrP} = k_{greenalite} \, e^{b_{greenalite} \, log_{10} SI_{greenalite}} \tag{31}$$

where $k_{greenalite}$ is $6.996 \times 10^{-13} \, M \, h^{-1}$ and $b_{greenalite}$ equals 1.856 (Table 3).

## 3.2 The sulphur cycle

The oxidised form of dissolved sulphur in the ocean is sulphate ($SO_4^{2-}$). Under anaerobic conditions, sulphate is used as terminal electron acceptor during organic matter remineralisation:

$$CH_2O + \frac{1}{2}SO_4^{2-} + H^+ \rightarrow CO_2 + \frac{1}{2}\Sigma H_2S + H_2O \tag{32}$$

The produced sulphide can then be re-oxidised when it comes into contact with oxygen via canonical sulphide oxidation (CSO)

$$\Sigma H_2S + 2O_2 \rightarrow SO_4^{2-} + 2H^+ \tag{33}$$

The rate of CSO in cGENIE is dependent on the concentration of the electron donor ($\Sigma H_2 S$) and acceptor ($O_2$) and a second order rate constant $k_{CSO} = 0.625 \times 10^6 \, M^{-2} \, h^{-1}$ (Zhang and Millero, 1993):

$$R_{CSO} = k_{CSO}[\Sigma H_2 S][O_2]^2 \tag{34}$$

Alternatively, $\Sigma H_2 S$ can be re-oxidised with $Fe^{3+}$ (either $Fe^{3+}_{total}$ or solid $FeOOH$) during sulphide-mediated iron reduc-
tion (as described in reaction 18).

In cGENIE, the eventual sink for sulphur is the precipitation of pyrite (reaction 21) (ignoring sulphide reacting with organic matter – see Hülse et al. (2019)). We do not currently include precipitation of gypsum ($CaSO_4$) in our model description (Fig. 4b). Gypsum is an evaporite mineral that precipitates during regional and episodic events of supersaturation, and was likely a less important sulphur sink on a globally integrated basis during Precambrian time, or during any other period in which ocean
$[SO_4^{2-}]$ was relatively low (Grotzinger and Kasting, 1993; Crowe et al., 2014; Fakhraee et al., 2019). Indeed, there is still some debate as to the time-integrated impact of sulfate evaporites on the steady-state global sulfur cycle even during more recent periods of Earth's history (Halevy et al., 2012; Canfield, 2013). However, due to its episodic nature, gypsum could play an important role as a sulphate source during transient events (for example during events of enhanced weathering of a gypsum-rich source; Shields et al., 2019). Planned future developments to cGENIE will incorporate an explicit gypsum cycle, which
would allow us to use the cGENIE.muffin model to investigate transient events of enhanced sulphate delivery to the ocean (see, e.g.; Shields et al., 2019).

## 3.3 Isotope geochemistry

A particularly important application of having a representation of an anoxic Fe-S cycle in cGENIE is in exploring ocean redox landscapes during Precambrian time, when ocean biogeochemical cycling was dominated by iron and sulphur (Raiswell and
Canfield, 2012). As noted above, one way of comparing our model output to available data is the explicit simulation of the burial phases of iron, which allows comparison to the often used Fe-proxy (Poulton and Canfield, 2011). A different and independent constraint potentially exists in the form of Fe or S isotopes (see e.g. Beard et al., 1999; Gomes and Johnston, 2017; van de Velde et al., 2018).

Any chemical element with multiple stable isotopes ($Fe$ for example has four stable forms $^{54}Fe$, $^{56}Fe$, $^{57}Fe$ and $^{58}Fe$)
can potentially be used to track physicochemical processes that act to partition stable isotopes according to thermodynamic or kinetic principles. For $Fe$ the most abundant isotopes are $^{54}Fe$ and $^{56}Fe$, and deviations in the $^{56}Fe/^{54}Fe$ ratio of Fe-bearing aqueous and mineral phases from that of a reference material can be described using conventional $\delta$-notation:

$$\delta^{56}Fe = \left( \frac{\left( \frac{^{56}Fe}{^{54}Fe} \right)}{\left( \frac{^{56}Fe}{^{54}Fe} \right)_{ref}} - 1.0 \right) \times 1000 \tag{35}$$

where $\left( \frac{^{56}Fe}{^{54}Fe} \right)_{ref}$ is the isotope ratio of a standard reference material (IRMM-14). Any geochemical reaction, be it biotic
(mediated by micro-organisms) or abiotic, can induce isotopic fractionation between co-occurring Fe- or S-bearing phases.

**Table 3.** List of kinetic constants for the reactions included in the cGENIE model. Note that the units are not identical to the units used in the user-configuration files of the cGENIE model. A table with the parameter units converted to cGEnIE units is included in Appendix A.

| Reactivity constants | Symbol | Unit | Value | Reference |
|---|---|---|---|---|
| Limitation constant oxygen reduction | $K_{0,O_2}$ | $M$ | $8.0 \times 10^{-6}$ | [1] |
| Inhibition constant oxygen reduction | $K_{i,O_2}$ | $M$ | $8.0 \times 10^{-6}$ | [1] |
| Limitation constant DIR | $K_{0,FeOOH}$ | $M$ | $1.0 \times 10^{-2}$ | [2] |
| Inhibition constant DIR | $K_{i,FeOOH}$ | $M$ | $1.0 \times 10^{-2}$ | [2] |
| Limitation constant DSR | $K_{0,SO_4^{2-}}$ | $M$ | $5.0 \times 10^{-4}$ | [3] |
| Inhibition constant DSR | $K_{i,SO_4^{2-}}$ | $M$ | $1.0 \times 10^{-3}$ | [1] |
| Canonical sulphide oxidation | $k_{CSO}$ | $M^{-2}h^{-1}$ | $0.625 \times 10^6$ | [4] |
| Ferrous iron oxidation | $k_{FIO}$ | $M^{-1}h^{-1}$ | $0.115 \times 10^6$ | [5] |
| Sulphide-mediated iron reduction (dissolved) | $k_{SMI_d}$ | $M^{-0.5}h^{-1}$ | $2.64 \times 10^2$ | [6] |
| Sulphide-mediated iron reduction (solid) | $k_{SMI_s}$ | $M^{-0.5}h^{-1}$ | $1.98 \times 10^0$ | [6] |
| $Fe_{free}^{3+}$ Scavenging constant | $k_{scav}$ | $mol^{-1}m^2$ | $1.43 \times 10^{-6}$ | [7] |
| $L-Fe^{3+}$ complex stability constant | $K_{sp}^{FeL}$ | $M^{-1}$ | $1.0 \times 10^{11}$ | [7] |
| Solubility product $FeS_{aq}$ | $K_{sp}^{FeS_{aq}}$ | $M$ | $8.32 \times 10^{-6}$ | [8] |
| Kinetic constant pyrite precipitation | $k_{PyP}$ | $M^{-1}h^{-1}$ | $0.3708 \times 10^0$ | [9] |
| Kinetic constant siderite precipitation | $k_{AFC}$ | $Mh^{-1}$ | $1.963 \times 10^{-14}$ | [10] |
| Kinetic exponent siderite precipitation | $b_{AFC}$ | - | $9.042 \times 10^0$ | [10] |
| Solubility product greenalite | $K_{sp}^{greenalite}$ | $M^{-1}$ | $3.98 \times 10^{27}$ | [11] |
| Kinetic constant greenalite precipitation | $k_{greenalite}$ | $Mh^{-1}$ | $6.996 \times 10^{-13}$ | [12] |
| Kinetic exponent greenalite precipitation | $b_{greenalite}$ | - | $1.856 \times 10^0$ | [12] |
| *Activity coefficients* | | | | |
| Activity constant $H^+$ | $\gamma_{H+}$ | - | 0.73 | [13] |
| Activity constant $OH^-$ | $\gamma_{OH-}$ | - | 0.69 | [14] |
| Activity constant $CO_3^{2-}$ | $\gamma_{CO_3^{2-}}$ | - | 1.17 | [15] |
| Activity constant $Fe^{2+}$ | $\gamma_{Fe2+}$ | - | 0.23 | [14] |
| Activity constant $SiO_2$ | $\gamma_{SiO_2}$ | - | 1.13 | [14] |

[1](Ridgwell et al., 2007),[2](Thamdrup, 2000),[3](Olson et al., 2016),[4](Zhang and Millero, 1993),[5](Millero et al., 1987a),[6](Poulton et al., 2004),[7](Ridgwell and DeAth, in prep.),[8](Luther et al., 1996),[9](Rickard, 1997),[10](Jiang and Tosca, 2019),[11](Tosca et al., 2015),[12](Rasmussen et al., 2015),[13](Marion et al., 2011),[14](Following the Davies equation),[15](Johnson , 1982)

To model the isotopic signatures of $Fe$ and $S$, we track the concentrations of the 'bulk' pools ($C_i$) and the isotope-specific pools ($^{56}C_i$ for example, in the case of $Fe$). The isotopic signature of an $Fe$ species $C_i$ is then calculated as

$$\delta^{56}Fe_{C_i} = \left( \frac{\left( \frac{^{56}C_i}{C_i - {}^{56}C_i} \right)}{\left( \frac{^{56}Fe}{^{54}Fe} \right)_{ref}} - 1.0 \right) \times 1000 \tag{36}$$

Each individual reaction $R_k$ is assigned a fractionation factor ${}^{56}\epsilon_{R_k}$ (in ‰; Table 4), which relates to ${}^{56}\alpha_{R_k}$ as

$$
{}^{56}\alpha_{R_k} = 1 + \frac{{}^{56}\epsilon_{R_k}}{1000} \tag{37}
$$

Isotope fractionation is then implemented by calculating an isotope specific reaction rate (for the ${}^{56}Fe$ pool) from the bulk reaction rate $R_k$

$$
{}^{56}R_k = \frac{{}^{56}\alpha_{R_k}\,{}^{56}r_{C_i}}{1 + {}^{56}\alpha_{R_k}\,{}^{56}r_{C_i}} R_k \tag{38}
$$

where

$$
{}^{56}r_{C_i} = \frac{{}^{56}C_i}{C_i - {}^{56}C_i} \tag{39}
$$

In this way, we assign a fractionation factor to each of the reactions considered in our model, and are able to track the isotopic signature of each $Fe$ and $S$ species. This allows us to simulate the $\delta^{56}Fe$ and $\delta^{34}S$ values of dissolved species and solid mineral phases, both of which can potentially be compared to observations from the geological record.

The obvious limitation of assigning a constant fractionation factor to each reaction is that this is incapable of fully capturing natural isotopic variability. For instance, different microbial strains of sulphur reducers (or iron oxidisers) express different fractionation factors, even though the overall reaction remains the same (Gomes and Johnston, 2017; Pellerin et al., 2019). This is reflected in the often broad range of isotopic fractionation factors found in the literature for a given process (Table 4). Other factors influencing microbial fractionation are local environmental conditions, such as electron donor type/availability (Wing and Halevy, 2014; Pellerin et al., 2018), or evolutionary adaptation (Pellerin et al., 2015). Aside from biologically mediated transformations, kinetic effects associated with abiotic aqueous reactions and precipitation of solid phases also affect the fractionation that is eventually recorded in the end-product. All these factors can make interpretation of isotope fractionations very complex in natural settings, and it is thus highly unlikely that any particular model simulation will be able to exactly reproduce observed isotope records. Nevertheless, the scheme employed here should be able to discern first-order observations from the geologic record, and in some cases could potentially be used to rule out particular end-member hypotheses for ocean chemistry.

## 4    Model testing

We evaluate our model in two ways. We first test whether our extended model code is able to reproduce the Fe cycle of the contemporary, oxygenated ocean, by comparing it with the previously validated standard (oxic) Fe cycle in cGEnIE (Tagliabue et al., 2016). Secondly, we assess how the model behaves in an ocean which is predominantly anoxic. For contrasting with observations, we are severely limited because the modern ocean is largely well-oxygenated, with only a few oxygen-minimum-zones near highly productive margins such as the Indian ocean and the Peruvian margin (Keeling et al., 2010). However, even in these regions dissolved oxygen concentrations rarely reach zero, and the development of ferruginous or euxinic conditions is essentially absent. Only highly restricted basins such as the Cariaco basin or the Black and Baltic Seas can develop euxinic

**Table 4.** Isotope fractionation factors

| Reaction | Reactant | Product | Fractionation factor | Literature min | Literature max | References |
|---|---|---|---|---|---|---|
| $^{34}\epsilon_{R-P}$ *(S fractionation)* | | | | | | |
| Dissimilatory sulphate reduction | $SO_4^{2-}$ | $H_2S$ | $-30.0‰$ | $-70.0‰$ | $-2.0‰$ | [1]-[3] |
| Canonical sulphide oxidation | $H_2S$ | $SO_4^{2-}$ | $-10.0‰$ | $-18.0‰$ | $+12.5‰$ | [4],[5] |
| Sulphide mediated iron reduction | $H_2S$ | $SO_4^{2-}$ | $-1.8‰$ | $-3.6‰$ | $0.0‰$ | [6],[7] |
| Pyrite precipitation | $H_2S$ | $FeS_2$ | $0.0‰$ | $-0.4‰$ | $1.2‰$ | [8],[9] |
| $^{56}\epsilon_{R-P}$ *(Fe fractionation)* | | | | | | |
| Dissimilatory iron reduction | $Fe^{3+}$ | $Fe^{2+}$ | $-1.3‰$ | $-2.95‰$ | $-1.3‰$ | [10]-[14] |
| Sulphide mediated iron reduction | $Fe^{3+}$ | $Fe^{2+}$ | $-1.3‰$ | $-1.3‰$ | $-1.3‰$ | [10]-[14] |
| Ferrous iron oxidation | $Fe^{2+}$ | $Fe^{3+}$ | $0.8‰$ | $0.4‰$ | $1.1‰$ | [13],[15] |
| Iron oxide precipitation[1] | $Fe^{3+}$ | $FeOOH$ | $0.0‰$ | - | - | - |
| Pyrite precipitation | $Fe^{2+}$ | $FeS_2$ | $-2.2‰$ | $-2.9‰$ | $-1.5‰$ | [15],[16] |
| Siderite precipitation | $Fe^{2+}$ | $FeCO_3$ | $-0.3‰$ | $-0.6‰$ | $0.0‰$ | [12],[17] |
| Greenalite precipitation[2] | $Fe^{2+}$ | $Fe_3Si_2O_5(OH)4$ | $0.0‰$ | - | - | - |

[1] (Kaplan and Rittenberg, 1964), [2] (Detmers et al., 2001), [3] (Sim et al., 2011), [4] (Gomes and Johnston, 2017), [5] (Pellerin et al., 2019), [6] (Poser et al., 2014), [7] (Fry et al., 1988), [8] (Bottcher et al., 1998), [9] (Price and Shieh, 1979), [10] (Beard et al., 1999), [11] (Beard et al., 2003), [12] (Johnson et al., 2004), [13] (Bullen et al., 2001),[14] (Crosby et al., 2007), [15] (Rolison et al., 2018), [16] (Guilbaud et al., 2011), [17] (Wiesli et al., 2004)

[1] The isotopic fractionation for iron oxide precipitation is driven by the oxidation reaction (FIO). [2] The isotope fraction factor for greenalite precipitation is currently unknown.

conditions, but these conditions arise as a result of local circulation within silled or enclosed basins, and are not likely to be representative of an anoxic open ocean setting. As a result, we lack observations to which we can directly calibrate our model. However, as our model development consisted of the implementation of well-established, mechanistic biogeochemical reactions, with relatively well-defined kinetic rates, our model should simulate realistic rates and concentrations without extensive calibration of model parameters. We illustrate this by showing the spatial concentration and isotope features of a hypothetical anoxic ocean (section 4.2), and – where possible – compare our predicted reaction rates to rates obtained from anoxic process analogues for the ancient oceans. Subsequently, we broadly illustrate the sensitivity of the model output to the newly introduced parameters of the iron-sulphur cycling (section 4.3). As discussed below, our model is largely robust across a range of values for key parameters, and predicts reaction and process rates that are comparable to those obtained experimentally or observed in modern analogue environments.

## 4.1 The contemporary ocean

We test our new iron scheme in comparison to the results of the oxic-only scheme presented by Tagliabue et al. (2016) which was based on the modern cGENIE configuration of Cao et al. (2009). For clarity, we carry out this test step-wise in 2 stages in

order to separate out the consequences of resolving the different oxidation states of dissolved iron, from the various solid-iron reactions:

1. Firstly, we substitute the original three tracers carried in the ocean – free dissolved iron (which we can equate here to $Fe^{3+}_{free}$), ligand-bound iron ($Fe^{3+}_{ligand}$), and free ligand ($L_{free}$), for our new tracer scheme of $Fe^{3+}$ (total = free + ligand-bound), $Fe^{2+}$, and TDL (total dissolved ligands). Note that in the original scheme only 2 tracers were actually needed – total dissolved iron ($Fe^{3+}$) and total dissolved ligand ($L$), as the equilibrium between $Fe^{3+}_{free}$, $Fe^{3+}_{ligand}$, and $L_{free}$ was re-calculated each time-step. Hence our new tracer scheme is these same 2 primary tracers ($Fe^{3+}$, $L$), plus $Fe^{2+}$. We then enable the 2-way oxidation/reduction reactions: $Fe^{2+}$ -> $Fe^{3+}$, and $Fe^{3+}$ -> $Fe^{2+}$ (Eqs. 13 and 18). We retain the same scavenging scheme as before, with iron scavenged by POM as $Fe^{3+}$ and not subject to DIR.

2. As per (1), but we now represent scavenged iron as $FeOOH$ and enable DIR (Eq. 15). We also enable SMI of solid $FeOOH$ with $SigmaH_2S$ (Eq. 18) and pyrite precipitation (Eq. 23). Note that the biogeochemical scheme used in Tagliabue et al. (2016) already included sulphate reduction and sulphide oxidation (DSR and CSO; Eqs. 32 and 33).

We find only minor differences between the old cGEnIE Fe cycling scheme (which only includes an oxic Fe cycle - see above) and the step 1 configuration of our new scheme (Fig. 6). Because the old scheme did not include reduced $Fe^{2+}$, we compare total dissolved Fe (TDFe), which is the sum of all dissolved Fe phases (i.e. [TDFe] = $[Fe^{3+}]$ + $[Fe^{2+}]$). Adding $Fe^{2+}$ as a tracer, which is released during POC oxidation under anoxic conditions, leads to a slight increase in [TDFe] in oxygen-minimum-zones (OMZ) (Fig. 6b), because in our new scheme, $Fe^{2+}$ is assumed not subject to scavenging. Hence reducing $Fe^{3+}$ to $Fe^{2+}$ reduces iron loss due to scavenging and enhanced TDFe in OMZs. Consequently, there is a slightly enhanced supply of Fe to the surface waters via upwelling, leading to marginally higher surface [TDFe] ($\sim$ 0.02 nM), in particular above the Peruvian OMZ (Fig. 6e). In step 2, by including a full Fe redox cycle (i.e. the reduction of FeOOH coupled to POC oxidation or sulphide oxidation) this effect is slightly intensified, with DIR and SMI leading to an addition reduction of $Fe^{3+}$ (as FeOOH) and release of $Fe^{2+}$ (which again, escapes scavenging). (Fig. 6c,f).

The small differences in [TDFe] translate to a negligible increase in global export production (Table 5) and indicates that the difference in performance between the old and new cGEnIE Fe cycle scheme are relatively trivial in a modern and near fully oxygenated ocean. For both the new as the old scheme, the sink of iron is scavenging of its oxidised form and no $FeS_2$ is formed (Table 5). Pyrite will only precipitate once FeS has passed its solubility threshold, which is $\sim$ 2 µM and such concentrations are not reached in the modern ocean. Because the new scheme is more mechanistic, as it allows for reduction of $Fe^{3+}$ in anoxic conditions, we believe our new extension will also be beneficial for studies of the modern iron cycle - in particular when addressing topics such as future ocean deoxygenation or intervals of the more recent geological past characterized by much more extensive OMZs than present..

## 4.2 An anoxic ocean

For testing and characterizing the Fe-S cycle model developments under anoxic conditions, we configure cGENIE with a single continent that runs from pole to pole (Fig. 2c; Fig. 3a). Each model experiment is initialised from a homogeneous and static

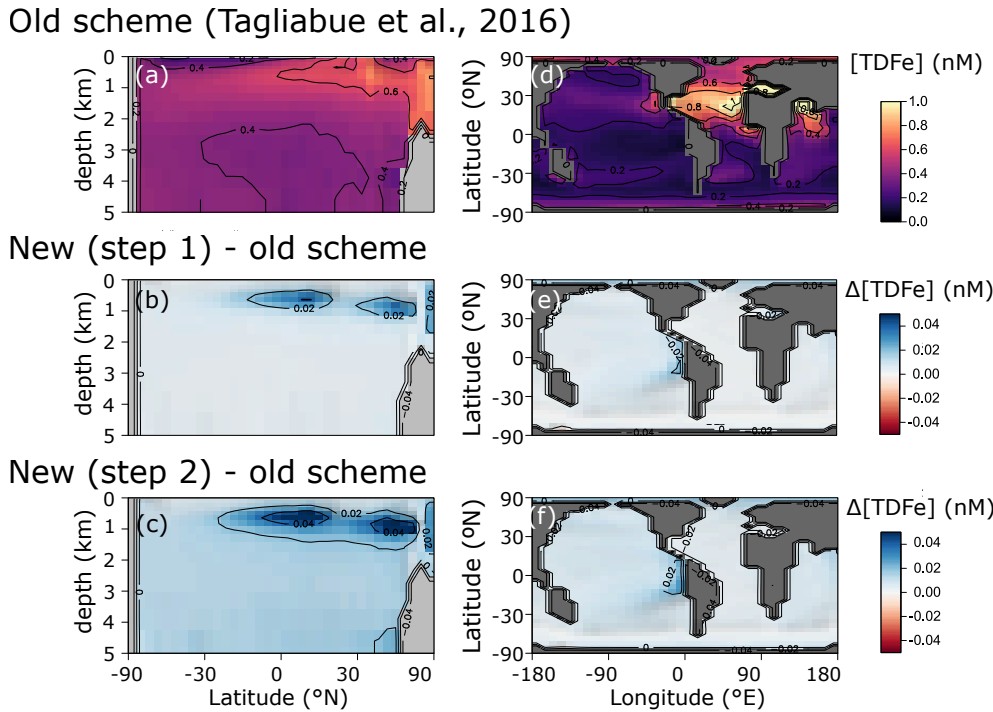

**Figure 6.** Comparison of total dissolved Fe concentrations ([TDFe] = $[Fe^{3+}]$ + $[Fe^{2+}]$) between (a,d) the old cGEnIE scheme (Tagliabue et al., 2016) and (b,e) the new scavenging scheme and (c,f) the full new scheme. (a-c) Zonally averaged TDFe concentrations, (d-f) surface TDFe concentrations. All units are in nM. (b,c,e,f) Positive values indicate the new scheme has higher concentrations, negative values indicate the old scheme has higher concentrations.

**Table 5.** Comparison of selected model output for the old Fe scheme and the new Fe scheme.

| Variable | Units | (Tagliabue et al., 2016) | This study, extension 1 | This study, extension 2 |
|---|---|---|---|---|
| mean $[O_2]$ | $\mu M$ | 166.61 | 166.373 | 166.135 |
| mean $[TDFe]$ | $nM$ | 0.455 | 0.456 | 0.472 |
| *POC* Export production | $Pg\ C\ yr^{-1}$ | 6.126 | 6.135 | 6.144 |
| $CaCO_3$ Export production | $Pg\ C\ yr^{-1}$ | 0.811 | 0.812 | 0.813 |
| *POC* flux to sediments | $Pg\ C\ yr^{-1}$ | 0.524 | 0.524 | 0.524 |
| $CaCO_3$ flux to sediments | $Pg\ C\ yr^{-1}$ | 0.450 | 0.450 | 0.451 |
| $FeOOH$ burial | $mol\ Fe\ yr^{-1}$ | $1.814 \times 10^9$ | $1.814 \times 10^9$ | $1.815 \times 10^9$ |
| $FeS_2$ burial | $mol\ Fe\ yr^{-1}$ | n.a. | n.a. | 0.0 |

ocean, with an imposed constant atmospheric $O_2$ concentration of 0.1 PAL (present atmospheric level, i.e. 21,000 ppm) and an atmospheric $CO_2$ concentration of $\sim$ 16 PAL (5337.6 ppm). These atmospheric boundary conditions are chosen as broadly plausible for the Precambrian Earth system and deliberately preclude the formation of sea-ice in order to simplify the resulting spatial tracer patterns of in the ocean and their mechanistic interpretation. The model is run in 'closed' configuration for all elements (notably $C$ and $P$), in which the ocean-atmosphere inventory for each element is always conserved. We chose to keep a closed configuration for $C$ and $P$, because this allows us to fix important boundary conditions (such as $pCO_2$ and productivity), and look at the emerging ocean redox state under these conditions. An exception is made for $Fe$ and $S$, which enter the surface ocean in the form of $Fe^{2+}$ and $SO_4^{2-}$, and exit the ocean as $FeOOH$, $FeS_2$, $FeCO_3$ or $Fe_3Si_2O_5(OH)_4$. Fluxes of $Fe$ and $S$ are chosen to be in balance with respect to $FeS_2$ burial (S:Fe=2:1), and to represent the best estimate of present-day weathering fluxes to the ocean (excluding reprocessing in inner shelf sediment settings) ($F_{Fe^{2+}} = 1.3 \times 10^{12}\ mol\ yr^{-1}$, $F_{SO_4^{2-}} = 2.6 \times 10^{12}\ mol\ yr^{-1}$; Poulton and Raiswell, 2002; Raiswell and Canfield, 2012). Hydrothermal systems represent another potentially important flux of $Fe$ to the ocean (Tagliabue et al., 2010; Conway and John, 2014; Lough et al., 2019), and this flux is likely to be elevated when ocean chemistry is pervasively anoxic and relatively low in $SO_4^{2-}$ (Kump and Seyfried, 2005). Our simulations therefore also include a hydrothermal flux of $Fe^{2+}$, broadly comparable to a plausible input flux to Proterozoic oceans ($15.1 \times 10^{12}\ mol\ yr^{-1}$; Thompson et al., 2019). This $Fe^{2+}$ flux in our model setup is equally distributed along a 'hydrothermal ridge' located in the middle of the ocean (a straight line from k=9, l=3 to k=9, l=15; Fig. 2a). In order to balance the S:Fe flux ratio at 2:1 (see above), we must also specify a uniform surface flux of $SO_4^{2-}$ of $30.2 \times 10^{12}\ mol\ yr^{-1}$. This is an extremely high S flux, and is unlikely to be realistic on long timescales. However, it allows us to quickly diagnose spatial patterns in reducing Fe-S cycling at steady state, without *a priori* introducing a bias towards ferruginous or euxinic redox states. We initialise the model with an average ligand concentration of 1 nM (compareable to the modern ocean) and a semi-arbitrary marine phosphate inventory that is 50% of today in order to reduce marine primary productivity (as productivity in the Precambrian ocean was likely lower than today; Crockford et al., 2018; Ozaki et al., 2019), and run the ocean circulation to steady state for 20,000 years for each individual experiment and present model output from the 20,000th year of integration. It is important to emphasise that the idealised configuration we implement here is not meant to represent any specific period or event in Earth's history, but is rather meant to serve as a broadly plausible and computationally efficient set of boundary conditions for testing the extended model code.

The sea surface temperature and ocean circulation generated by our configuration of cGENIE are shown in Fig. 3. Sea surface temperatures vary from $\sim$40 °C at the equator to $< 10$ °C at the poles (Fig. 3c). The barotropic stream function shows a large degree of symmetry (Fig. 3d), whereas the overturning patterns are skewed to the southern hemisphere, with a strong anticlockwise circulation at around -60 °N (Fig. 3e). The overturning stream function shows strong upwelling at the equator and deep-water mixing at the poles (Fig. 3e). The wind stress at the equator drives surface waters towards the west, which will lead to stronger upwelling on the west side of the continent. The redox patterns discussed below will reflect these two main features: (i) upwelling at the equator, in particular at the west side of the continent and (ii) deep-water mixing at the poles.

In the following section, we briefly discuss the spatial model output and modelled redox cycling for our baseline configuration. Precipitation fluxes of oxidised $Fe^{3+}$ are specified according to the scavenging scheme (Eq. 11). We focus here on:

(i) three depth slices of particle sinking fluxes of $POC$, $FeOOH$, $FeS_2$, and $Fe_3Si_2O_5(OH)_4$ (Fig. 7; $FeCO_3$ fluxes were negligible and are not shown); (ii) the average and three vertical longitudinal slices of $O_2$, dissolved $Fe^{3+}$, $Fe^{2+}$ and dissolved $\Sigma H_2S$ (Fig. 8); and (iii) an overview of globally averaged reaction rates and iron mineral burial fluxes (Table 6).

The $POC$ flux decreases with depth in the model from a maximum of $\sim 10\ mmol\ C\ m^{-2}\ d^{-1}$ immediately below the surface layer, to near-zero values at the seafloor (Fig. 7a-c). The spatial pattern of solid $FeOOH$ flux (following the scavenging scheme described earlier – see section 3.1.1) matches the $POC$ flux pattern immediately below the surface layer, with higher values at the equator and poles (dark blue shading in Fig. 7d). Most of this $FeOOH$ is reduced in anoxic subsurface layers and the flux declines with depth. However, at the poles where deep convection allows for greater oxygen (and $Fe^{3+}$) penetration into the ocean interior (Fig. 8a-h), sinking particles have more time to continue to scavenge and accumulate $Fe^{3+}$ and the flux increases with depth (Fig. 7e,f). The maximum $FeOOH$ flux at the seafloor reaches $\sim 300\ \mu mol\ m^{-2}\ d^{-1}$ (Fig. 7f), comparable to that observed in typical modern ocean margin sediments (see e.g.; van de Velde and Meysman, 2016). Fluxes of $FeS_2$ reach their maximum near the seafloor (where a source of deep $Fe^{2+}$ is supplied via the imposed hydrothermal flux), with maximum fluxes similar to those of $FeOOH$ (Fig. 7g-i). The spatial pattern of $Fe_3Si_2O_5(OH)_4$ formation is very similar to $FeS_2$, but the fluxes are several orders of magnitude lower (Fig. 7j-l). Fluxes of $FeCO_3$ where near-zero everywhere (data not shown), consistent with recent work suggesting that water column precipitation of $FeCO_3$ is difficult to achieve, even in iron-dominated oceans (Jiang and Tosca, 2019; Tosca et al., 2019). The negligible $FeCO_3$ fluxes are at odds with $Fe$-speciation data of Precambrian rocks that show an important fraction of sedimentary $Fe$ consists of reduced non-sulphurised $Fe$ minerals (Sperling et al., 2015). Our results indicate that these minerals are most likely formed during sedimentary diagenesis, emphasising the potential importance of processes below the sediment-water interface in structuring Fe-speciation signals in Earth's rock record. Future development will thus include a representation of sedimentary $Fe$ cycling in the sedimentary module 'OMEN-SED' (Hülse et al., 2018).

We additionally find that even in an idealised ocean with a simple symmetrical continental configuration, complex spatial patterns emerge in the Fe-S redox chemistry (Fig. 8). For instance, the poles are more well-ventilated, allowing oxygenated conditions and persistence of $Fe^{3+}$ to a few kilometres depth in our benchmark simulation (Fig. 8a-h). The concentrations of $Fe^{3+}$ are much higher than the nano - picomolar concentrations we observe in the ocean today (Tagliabue et al., 2016) – a consequence of much higher rates of iron delivery to the surface ocean (through upwelling of reduced $Fe^{2+}$), which allows $Fe^{3+}$ to accumulate to higher concentrations before it is eventually scavenged. Our model predicts concentrations in the hundreds of nanomolar range (up to micromolar at the oxic/anoxic interface), which compare well to oxic water layers overlying anoxic deep water (Taillefert and Gaillard, 2002; Crowe et al., 2008). Note that this $Fe^{3+}$ is likely in colloidal or nanoparticulate form and not truly dissolved. At eastward latitudes, deep convection at the poles is less intense, and upwelling on the eastward edge of the ocean leads to higher export production, which subsequently leads to build-up of reduced $Fe^{2+}$ and $\Sigma H_2S$ (Fig. 8l,p). Dissolved $Fe^{2+}$ reaches higher concentrations in the deeper ocean, largely as a result of deep hydrothermal inputs (Fig. 8i-l), whereas the highest $\Sigma H_2S$ concentrations are spatially constrained to to areas of more intense $POC$ degradation along the equator, just below the oxic zone (Fig. 8m-p).

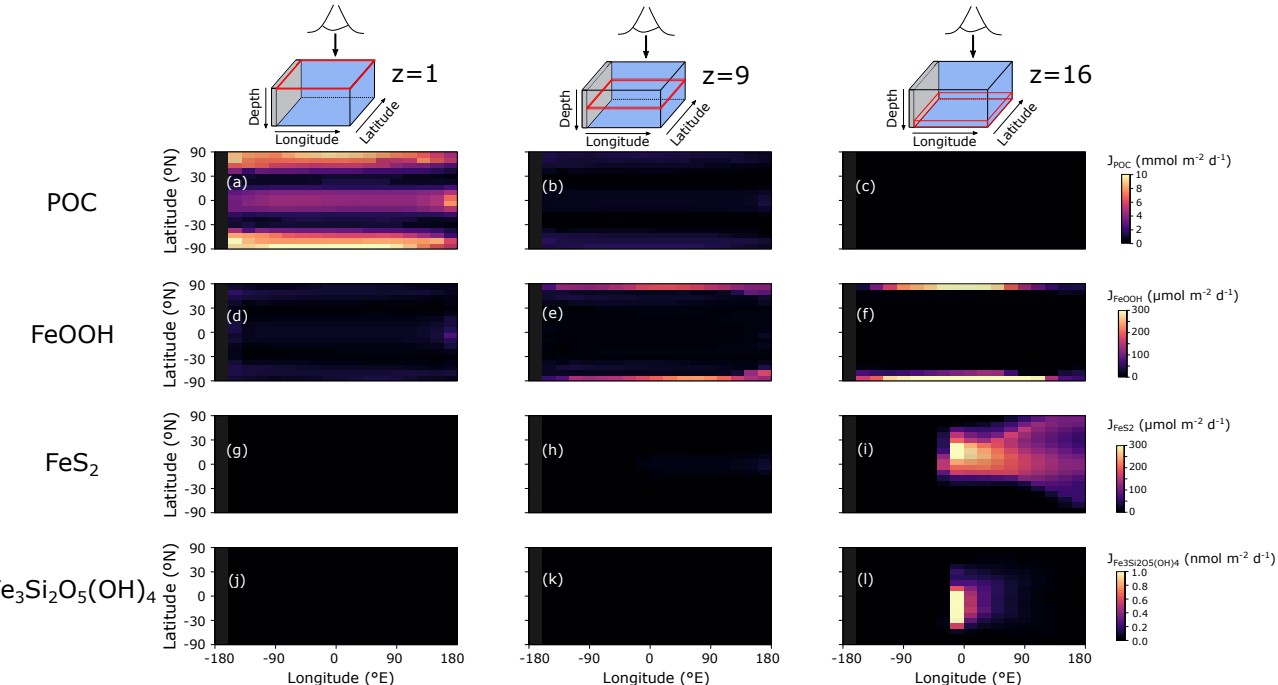

**Figure 7.** Model output of (a)-(c) Particulate organic carbon flux ($J_{POC}$), (d)-(f) solid iron oxide flux ($J_{FeOOH}$), (g)-(i) pyrite flux ($J_{FeS_2}$) and (j)-(l) greenalite flux ($J_{Fe_3Si_2O_5(OH)_4}$) for an ocean with 50 % of the modern phosphate inventory. $J_{POC}$ is given in $mmol\ m^{-2}\ d^{-1}$, $J_{FeOOH}$ and $J_{FeS_2}$ are given in $\mu mol\ m^{-2}\ d^{-1}$ and $J_{Fe_3Si_2O_5(OH)_4}$ is given in $nmol\ m^{-2}\ d^{-1}$. The siderite flux ($J_{FeCO_3}$) was near-zero everywhere and is not shown here.

Because pervasive anoxia is not present in modern open ocean environments (see above), evaluation of the realism of our globally integrated reaction rates must rely on comparison to modern process analogues for ancient oceans (e.g., anoxic lake and restricted marine systems). Though we consider this a valid approach, it must be borne in mind that the transport processes in particular are very different in stratified lacustrine and marine systems, and there are reasons to assume that the biogeochemical dynamics of these systems will not strictly map onto pervasively anoxic open ocean environments. However, though the absolute values are not directly comparable, we can qualitatively compare our results with rates derived from anoxic lake systems. For instance, our default model suggests that DIR is a negligible contributor to POC mineralisation (Table 6), consistent with previous observations from ferruginous Lake Matano and the euxinic Black Sea (Konovalov et al., 2006; Crowe et al., 2011). However, Crowe et al. (2011) found that rates of DIR were roughly an order magnitude lower than those of methanogenesis, whereas we find that DIR is 5 orders of magnitude less important than all other mineralisation pathways (Table 6). One possible reason for this discrepancy is the importance of sediment recycling in the natural lake system, which our model does not currently represent. Due to the shallower water column (e.g., $\sim$ 300m in Lake Matano versus 5000m in our idealized ocean), there is a much stronger coupling between sedimentary processes (i.e., the recycling of Fe as a benthic flux) and water column processes. Indeed, when the total rate of DIR is corrected for iron recycling occurring in sediments

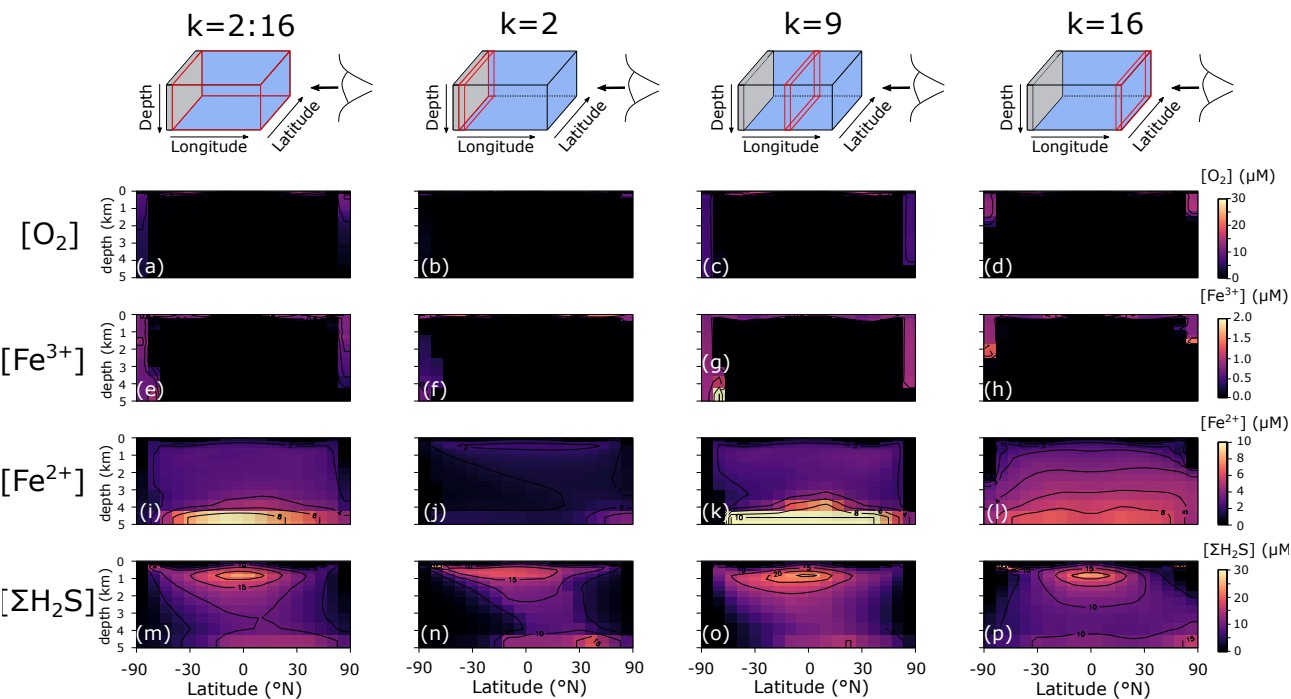

**Figure 8.** Model output of (a)-(d) dissolved oxygen ($O_2$), (e)-(h) dissolved ferric iron ($Fe^{3+}$), (i)-(l) total dissolved ferrous iron ($Fe^{2+}$) and (m)-(p) total dissolved sulphide ($\Sigma H_2 S$) for an ocean with 50 % of the modern phosphate inventory. All concentrations are in $\mu M$.

within Lake Matano the importance of DIR decreases by several orders of magnitude (Crowe et al., 2008). Additionally, cGENIE treats POC as a concentration assumed uniformly dispersed throughout a grid cell, meaning that POC and FeOOH concentrators used in reaction calculation will not take into account the reality of particulate matter being locally aggregated and highly concentrated as it sinks.

5    Our model predicts a globally integrated FIO rate of $\sim 0.86 \times 10^{-7} \, mol \, Fe \, kg^{-1} \, yr^{-1}$ (Table 6), lower than rates estimated for anoxic lakes ($12 - 51 \times 10^{-6} \, mol \, Fe \, kg^{-1} \, yr^{-1}$; Crowe et al., 2008; Walter et al., 2014). However, rates measured near the oxycline ($5 \times 10^{-6} \, mol \, Fe \, kg^{-1} \, yr^{-1}$) are comparable to rates measured in anoxic lake systems. Interestingly, more than half of the $\Sigma H_2 S$ is re-oxidised via SMI (Table 6), which indicates that $Fe$ is able to act as a relatively efficient intermediate between $\Sigma H_2 S$ oxidation and $O_2$ reduction. This also occurs in the Black Sea, where metal oxides are responsible for $\sim 60$ %
10    of the sulphide re-oxidation (Konovalov et al., 2006).

#### 4.2.1    Isotope patterns

Figure 9a-e shows the modelled stable Fe isotope patterns for all key Fe-bearing dissolved and solid-phase species (with the exception of $FeCO_3$, which is a negligible component in our benchmark simulation). In our model simulations, all dissolved $Fe$ that enters the ocean is assigned an isotope signature of 0.0 ‰. This allows us to observe the isotope fractionation of all $Fe$

**Table 6.** Globally integrated reaction rates and burial fluxes for our benchmark simulation.

| Reaction number | Reaction name | Units | Rate |
|---|---|---|---|
| R1 | Aerobic respiration | $mol\ POC\ kg^{-1}\ yr^{-1}$ | $0.88 \times 10^{-7}$ |
| R2 | Dissimilatory iron reduction | $mol\ POC\ kg^{-1}\ yr^{-1}$ | $0.96 \times 10^{-12}$ |
| R3 | Dissimilatory sulphate reduction | $mol\ POC\ kg^{-1}\ yr^{-1}$ | $0.69 \times 10^{-7}$ |
| R4 | Methanogenesis | $mol\ POC\ kg^{-1}\ yr^{-1}$ | $0.21 \times 10^{-6}$ |
| R5 | Ferrous iron oxidation | $mol\ Fe\ kg^{-1}\ yr^{-1}$ | $0.86 \times 10^{-7}$ |
| R6 | Canonical sulphide oxidation | $mol\ S\ kg^{-1}\ yr^{-1}$ | $0.60 \times 10^{-7}$ |
| R7a | Sulphide-mediated iron reduction (dissolved) | $mol\ S\ kg^{-1}\ yr^{-1}$ | $0.10 \times 10^{-7}$ |
| R7b | Sulphide-mediated iron reduction (solid) | $mol\ S\ kg^{-1}\ yr^{-1}$ | $0.16 \times 10^{-6}$ |
| R8 | Iron oxide precipitation | $mol\ FeOOH\ kg^{-1}\ yr^{-1}$ | $0.16 \times 10^{-6}$ |
| R9 | Pyrite precipitation | $mol\ FeS_2\ kg^{-1}\ yr^{-1}$ | $0.42 \times 10^{-8}$ |
| R10 | Siderite precipitation | $mol\ FeCO_3\ kg^{-1}\ yr^{-1}$ | $0.10 \times 10^{-62}$ |
| R11 | Greenalite precipitation | $mol\ Fe_3Si_2O_5(OH)_4\ kg^{-1}\ yr^{-1}$ | $0.15 \times 10^{-13}$ |
| - | Iron oxide burial | $mol\ FeOOH\ m^{-2}\ yr^{-1}$ | $0.13 \times 10^{-1}$ |
| - | Pyrite burial | $mol\ FeS_2\ m^{-2}\ yr^{-1}$ | $0.22 \times 10^{-1}$ |
| - | Siderite burial | $mol\ FeCO_3\ m^{-2}\ yr^{-1}$ | $0.54 \times 10^{-56}$ |
| - | Greenalite burial | $mol\ Fe_3Si_2O_5(OH)_4\ m^{-2}\ yr^{-1}$ | $0.25 \times 10^{-7}$ |

phases relative to the $Fe$ that entered the ocean (Fig. 10a). The dissolved iron phases show similar isotopic signatures ($\sim 1.1$ ‰; Fig. 9a,b; Fig. 10a), which can be explained by the large amount of isotopically light $FeS_2$ burial (Fig. 10a), which drives $\delta^{56}Fe - Fe^{2+}$ to heavier values. Note that the very heavy $\delta^{56}Fe - Fe^{3+}$ values only occur in the ocean interior, where the concentration of $Fe^{3+}$ is virtually zero (Fig. 8e-h). The isotope signature of buried oxidised iron ($FeOOH$) is around 1.8 ‰

5   heavier than the $Fe$ that entered the ocean, whereas the major burial fraction of reduced iron ($FeS_2$) has an isotope signature that is $\sim 1$ ‰ lighter, which reflects their relative importance as an Fe burial phase (Fig. 9c,e; Fig. 10a). These isotope values are broadly comparable to phase-specific stable Fe isotope observations from ancient sedimentary rocks (Heard and Dauphas, 2020). Similarly, Fig. 9f-g shows the modelled stable S isotope patterns for all key S-bearing dissolved and solid-phase species.

All dissolved $S$ that enters the ocean is assigned an isotope signature of 0.0 ‰. Sulphate is isotopically enriched relative

10   to its input value ($\sim 3.5$ ‰; Fig. 9f; Fig. 10b), and this difference compares well to the geological record ($SO_4^{2-}$ is $\sim 5$ ‰ heavier than its input value; Canfield and Farquhar, 2009). Additionally, free sulphide is isotopically lighter than sulphate ($\sim$ -5 ‰ on a global scale; up to $\sim$ -15 ‰ locally), while buried $FeS_2$ expresses an isotope signature of $\sim$ -2.38 ‰ (Fig. 10b). Our baseline simulation thus suggests that the expressed isotope fractionation between $SO_4^{2-}$ and $FeS_2$ is only around 6.5 ‰, which is roughly consistent with what is observed in the geological record (Canfield and Farquhar, 2009). Therefore, we

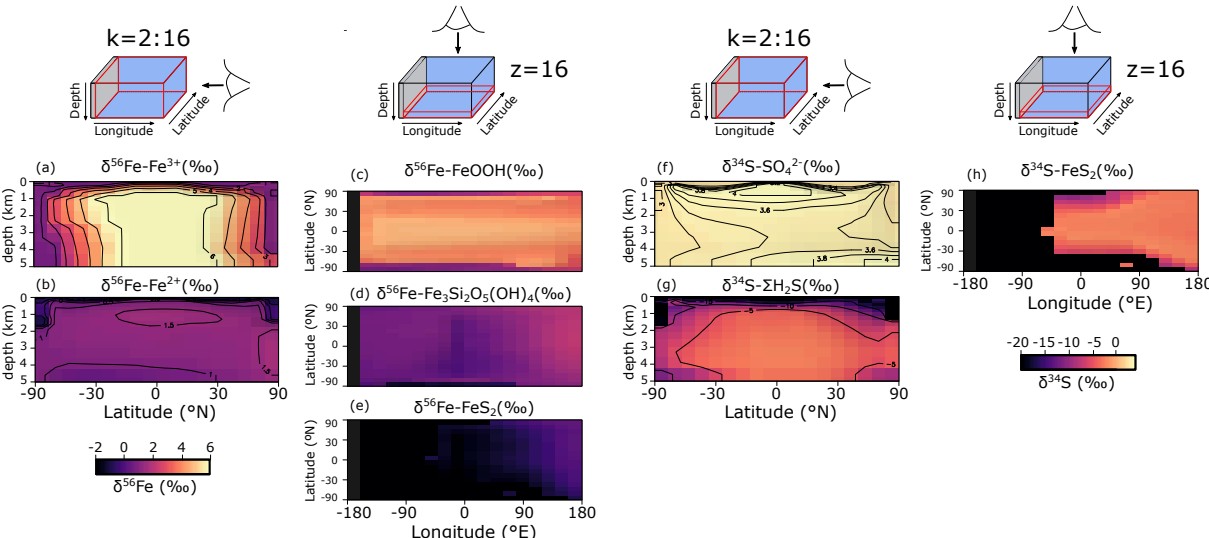

**Figure 9.** (a-e) Modelled stable Fe isotope signatures of key dissolved and solid-phase Fe species. Shown are zonally averaged values for (a) dissolved ferric Fe ($\delta^{56}Fe - Fe^{3+}$) and (b) dissolved ferrous iron ($\delta^{56}Fe - Fe^{2+}$), and the isotope compositions at the seafloor of (c) iron oxides ($\delta^{56}Fe - FeOOH$), (d) greenalite ($\delta^{56}Fe - Fe_3Si_2O_5(OH)_4$) and (e) pyrite ($\delta^{56}Fe - FeS_2$) for an ocean with 50 % of the modern phosphate inventory. All values are in ‰, relative to IRMM-14. (f-g) Modeled stable S isotope signatures of key dissolved and solid-phase S species. Shown are zonally averaged values for (f) sulphate ($\delta^{34}S - SO_4^{2-}$) and (g) dissolved free sulphide ($\delta^{34}S - \Sigma H_2S$), and (h) the isotope composition at the seafloor of pyrite ($\delta^{34}S - FeS_2$). All values are in ‰, relative to VCDT.

conclude that our model has strong potential for tracking iron and sulphur isotope signatures for comparison with Earth's rock record.

## 4.3 Sensitivity analysis

We evaluate model output sensitivity to four key parameters of our modelled iron-sulphur cycle ($K_{0,FeOOH}$, $K_{i,FeOOH}$, $k_{SMI,s}$, $k_{PyP}$) using the Elementary Effect Test (EET; Morris, 1991). These parameters were chosen because they are either unconstrained by laboratory experiments ($K_{0,FeOOH}$, $K_{i,FeOOH}$; Section 3.1.2), represent a complex mixture of different iron minerals with different reactivities ($k_{SMI,s}$; Poulton et al., 2004) or are expected to have a strong and potentially difficult to forecast influence on other reactions ($k_{PyP}$; van de Velde et al., 2020b). Other parameters introduced in the model are either relatively well-constrained by laboratory studies and calibrated on field data ($k_{CSO}$, $k_{FIO}$; Millero et al., 1987a, b; Ridgwell et al., 2007), have been calibrated extensively in previous work ($k_{scav}$; Tagliabue et al., 2016) or are likely to be of secondary importance to the iron-sulphur cycle ($k_{AFC}$, $b_{AFC}$, $k_{greenalite}$, $b_{greenalite}$).

(a) Fe mass balance

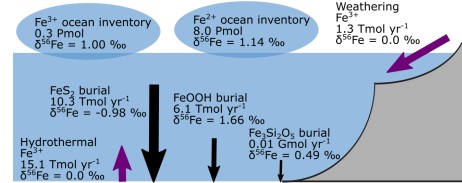

(b) S mass balance

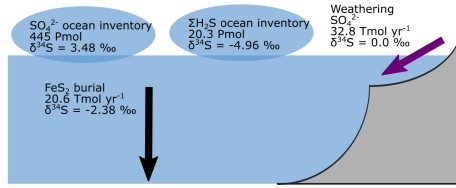

**Figure 10.** Isotope mass balance for the (a) $^{56}Fe$ and (b) $^{34}S$ systems from our baseline simulation. Purple arrows are influxes, black arrows outfluxes. Bubbles represent the whole ocean inventory. Note that in our simulation, the S system is not in steady-state (because $FeS_2$ burial is too low).

The EET method estimates global sensitivity by calculating the mean of $r$ finite differences ('Elementary Effects') (Pianosi et al., 2016),

$$
\begin{aligned}
S_i &= \frac{1}{r} \sum_{j=1}^{r} EE^j \\
&= \frac{1}{r} \sum_{j=1}^{r} \frac{f(\bar{x}_1^j, ..., \bar{x}_i^j + \Delta_i^j, ..., \bar{x}_M^j) - f(\bar{x}_1^j, ..., \bar{x}_i^j, ..., \bar{x}_M^j)}{\Delta_i^j} c_i
\end{aligned}
\tag{40}
$$

where $x_i^j$ represents the $j^{th}$ value of the $i^{th}$ parameter, $\Delta_i^j$ the variation on the $i^{th}$ parameter, $f()$ is the model output for a given set of parameters and $c_i$ is a scaling factor. A higher mean value $S_i$ indicates that a given model output is more sensitive to variations in parameter i. The standard deviation can also be calculated, with a high standard deviation indicating that a parameter interacts with others because its sensitivity changes across the variability space (inlay in Fig. 11; Pianosi et al., 2016).

We use the EET method, as implemented within the Sensitivity Analysis For Everyone (SAFE) toolbox (Pianosi et al., 2015), to investigate the four chosen model parameters across the ranges specified in Table 7. We vary the limitation and inhibition constants $K_{0,FeOOH}$ and $K_{i,FeOOH}$ over 6 orders of magnitude around the default/baseline value due to the high uncertainty associated with these parameters. The lower-bound of the reaction constant of solid iron oxide with sulphide ($k_{SMI,s}$) is defined as the reactivity of hematite ($5.34 \, 10^{-3} \, M^{-0.5}h^{-1}$; Poulton et al., 2004), and the upper bound is taken to be 5 orders of magnitude higher ($5.34 \, 10^2 \, M^{-0.5}h^{-1}$, which is comparable to freshly precipitated hydrous ferric oxide). We test the pyrite precipitation constant ($k_{Pyp}$) across a range between $0.3708 \, 10^{-3} \, M^{-1}h^{-1}$ and $0.3708 \, 10^2 \, M^{-1}h^{-1}$, which corresponds to the range of kinetic constants commonly used in diagenetic models (see e.g.; Van Cappellen and Wang,

**Table 7.** Sensitivity range and baseline values of each parameter tested. Note that the units are not identical to the units used in the user-configuration files of the cGENIE model. A table with the parameter units converted to cGEnIE units is included in Appendix A.

| Constant | Symbol | Unit | Baseline | Minimum | Maximum |
|---|---|---|---|---|---|
| Limitation constant DIR | $K_{0,FeOOH}$ | $M$ | $1.0 \times 10^{-2}$ | $1.0 \times 10^{-5}$ | $1.0 \times 10^{1}$ |
| Inhibition constant DIR | $K_{i,FeOOH}$ | $M$ | $1.0 \times 10^{-2}$ | $1.0 \times 10^{-5}$ | $1.0 \times 10^{1}$ |
| Sulphide-mediated iron reduction (solid) | $k_{SMI,s}$ | $M^{-0.5}h^{-1}$ | $1.98 \times 10^{0}$ | $5.34 \times 10^{-3}$ | $5.34 \times 10^{2}$ |
| Pyrite precipitation | $k_{PyP}$ | $M^{-1}h^{-1}$ | $0.3708 \times 10^{0}$ | $0.3708 \times 10^{-3}$ | $0.3708 \times 10^{2}$ |

1996; Meysman et al., 2003; Dale et al., 2009; van de Velde et al., 2020a). Our sensitivity ensemble consists of one hundred individual model experiments, using Latin-Hypercube sampling approach (using the SAFE toolbox; Pianosi et al., 2015) to select random starting points $x^j (j = 1, ...; r)$ and parameter variations $\Delta_i$. For more information on the sampling strategy we refer the interested reader to Campolongo et al. (2011).

The EET analysis suggests that changes in $K_{0,FeOOH}$ can significantly impact DIR, whereas the inhibition constant $K_{i,FeOOH}$ has a relatively minor impact on model output (Fig. 11). This is expected, as both parameters act only on the mineralisation pathways (Eq. 10). It is also consistent with some of the literature on anoxic systems, which suggests that in many cases the majority of iron reduction in the water column is coupled to sulphide (Mikucki et al., 2009). Even though both $K_{0,FeOOH}$ and $K_{i,FeOOH}$ are the least constrained by experimental results (see section 3.1.2), the EET analysis indicates that they are

relatively unimportant for the overall model output, despite their impact on DIR. In contrast, $k_{PyP}$ and $k_{SMI,s}$ both exerted more notable impact on model output across a range of diagnostics (Fig. 11). In particular, $k_{PyP}$ had a significant impact across all model output diagnostics analysed here both in terms of model sensitivity and interactivity with other parameters (Fig. 11). Because pyrite precipitation controls the inventories of both dissolved $Fe^{2+}$ and dissolved $\Sigma H_2 S$, reducing or increasing the kinetic precipitation parameter will affect the ambient concentrations, re-oxidation pathways, and eventual mineral burial for

all phases across the Fe-S system.

      Unfortunately, an Elementary Effect Test gives no quantitative metric to evaluate the magnitude with which a parameter affects overall model outcome. Therefore, to illustrate the quantitative impact of the possible parameter choices, we ran a separate set of experiments where we changed a parameter from its baseline value to its lower and upper bound, whilst keeping the other 3 parameters at their baseline values (Table 7).

Figure 12 reveals that AR, DSR, mean surface $O_2$ concentrations and CSO are relatively insensitive to parameter choices (Fig. 12a,c,d,i). Dissimilatory iron reduction is only important when $K_{0,FeOOH}$ is set at its lowest value, and even then it is three orders of magnitude smaller than AR or SR (Fig. 12a-c). Consistent with the EET analysis (Fig. 11), $K_{0,FeOOH}$ and $K_{i,FeOOH}$ have no influence on the model output, aside from the magnitude of DIR, which in itself is of less importance as mineralisation pathway (Fig. 12b and Section 4.2). In contrast, changes in the pyrite precipitation constant $k_{PyP}$ impact several

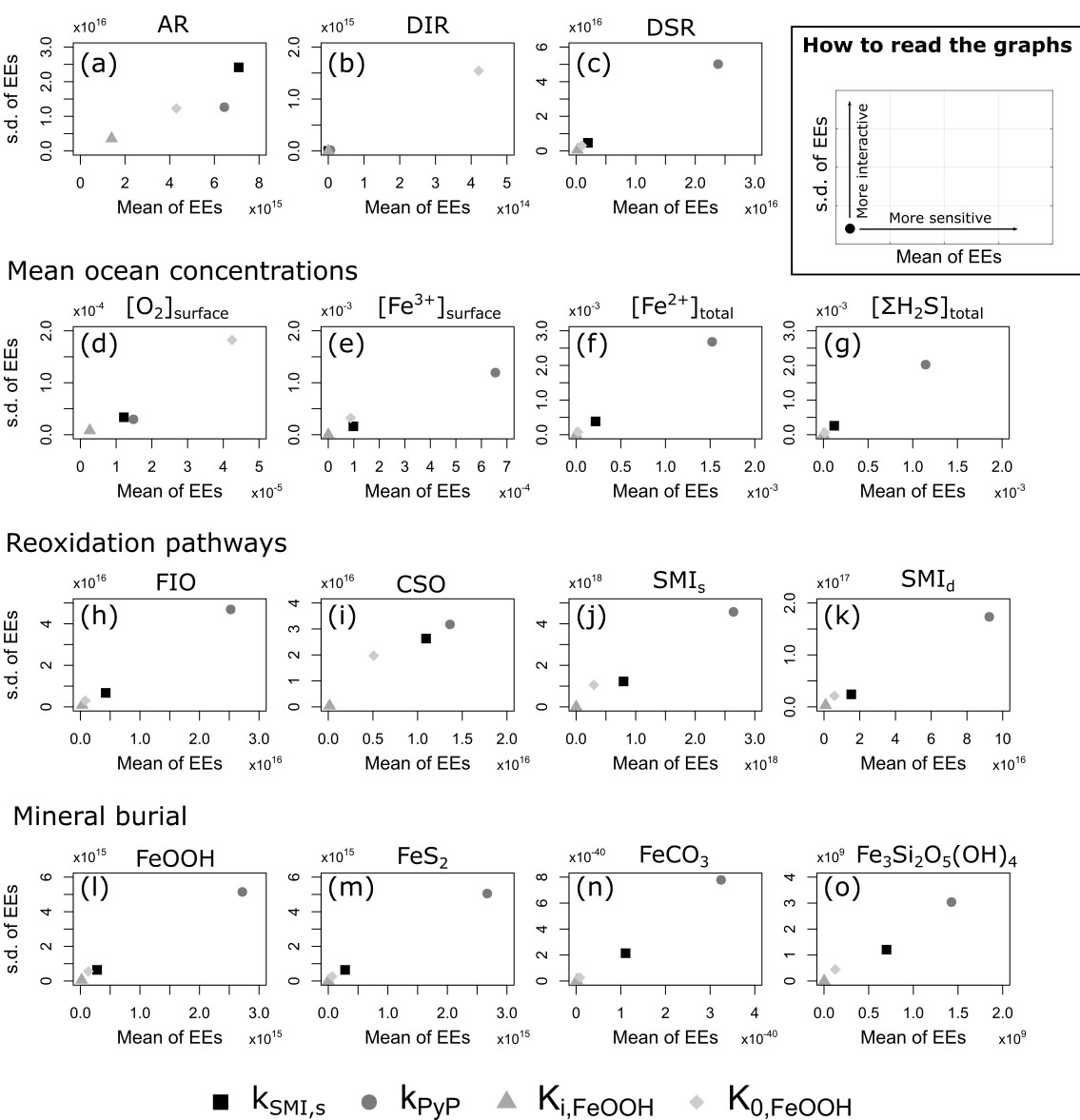

**Figure 11.** Sensitivity analysis of the four key-parameters of the iron-sulphur cycle, for a range of different outputs. Inset shows how to read the graph; points that plot more to the right indicate that the specific output is more sensitive to changes in parameter values, points that plot higher indicate that the parameter is more interactive with other parameters. Data processing was done with the SAFER toolbox of Pianosi et al. (2015).

model outputs. When pyrite precipitation rates are elevated, $Fe^{2+}$ and $\Sigma H_2 S$ are removed from local seawater more rapidly, which results in:

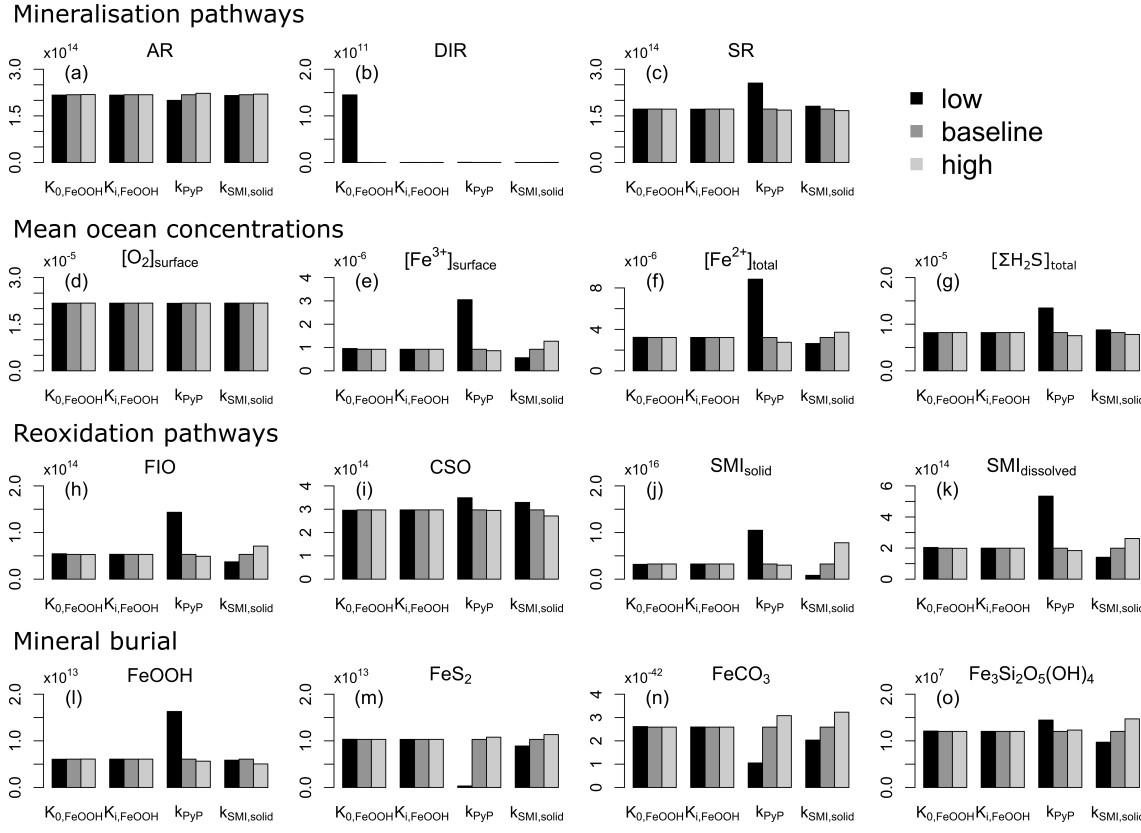

**Figure 12.** Global model output for the three different values of the four key parameters ($K_{0,FeOOH}$, $K_{i,FeOOH}$, $k_{PyP}$ and $k_{SMI,s}$). Values for low, baseline and high values are given in Table 7. Panels (a)-(c) and (h)-(n) are in $mol\ yr^{-1}$, panels (d)-(g) are in $mol\ kg^{-1}$.

1. a decreased build-up of $Fe^{2+}$ and $\Sigma H_2 S$ in the ocean interior (Fig. 12f,g), leading to

2. a decrease in both aerobic and anaerobic re-oxidation pathways (FIO, $SMI_d$ and $SMI_s$; Fig. 12h-k), which then results in

3. more $O_2$ available for aerobic respiration at the expense of less thermodynamically favourable electron acceptors (i.e. AR increases and DSR decreases; Fig. 12a,c) and

4. a decrease in $Fe^{3+}$ concentrations in the surface ocean (Fig. 12e), and

5. less burial of $FeOOH$, increasing the burial of reduced iron minerals (Fig. 12l-n)

The effect on greenalite burial is non-linear (Fig. 12o), which indicates that at higher values of $k_{PyP}$, pyrite precipitation is competing with greenalite precipitation. Overall, our sensitivity analysis suggests that $k_{PyP}$ is an important parameter for the model output, and should be chosen with care. Fortunately, pyrite precipitation has been well studied in laboratory experiments (Rickard, 1997, 2006), with the result that our baseline value for this parameter is relatively well constrained.

The fourth parameter, which also influences the model output, is the reactivity parameter of solid iron oxides ($k_{SMI,s}$). Here, the parameter choice is more complex. Laboratory experiments have shown that different iron oxide minerals exhibit a wide reactivity range (spanning several orders of magnitude) (Canfield, 1992; Poulton et al., 2004). Therefore, we explore the sensitivity of this parameter in more detail using a range of measured reactivity constants by Poulton et al. (2004), whilst keeping all other parameters at their baseline values (Fig. 13). Increasing the reactivity of particulate $FeOOH$

1. increases the anaerobic re-oxidation reaction of sulphide with $FeOOH$ (SMI$_s$) at the expense of the aerobic re-oxidation reaction (CSO) (Fig. 13i,j), which then leads to

2. an increase in the $Fe^{2+}$ inventory and a decrease in the $\Sigma H_2 S$ inventory (Fig. 13f,g), more $Fe^{2+}$ leads to

3. more FIO, and thus a higher surface $Fe^{3+}$ concentration, and more re-oxidation of $\Sigma H_2 S$ with dissolved $Fe^{3+}$ (SMI$_d$; Fig. 13e,h,k)

4. Because of the higher reactivity of the $FeOOH$ particles, less $FeOOH$ is buried, and more reduced $Fe$-minerals are buried (Fig. 13l-o)

Although it is clear that changing $k_{SMI,s}$ impacts model output, the overall magnitude of the effect is moderate when compared to changing $k_{PyP}$. Nevertheless, the choice of $k_{SMI,s}$ is critical. We choose a baseline reactivity value (Table 3) comparable to lepidocrocite for several reasons. Firstly, we assume all $Fe^{3+}$ that is not scavenged represents a 'colloidal' pool, with a reactivity of similar to that of hydrous ferric oxide. When $Fe^{3+}$ becomes scavenged (and is thus in solid state), it has likely undergone some ageing, and it will be less reactive than hydrous ferric oxide. Secondly, any $FeOOH$ that does not react in the water column will end up in the sediment, and will, at least in part, be recycled back to the water column (even under oxic conditions; Dale et al., 2015). Our model currently lacks a sedimentary iron cycle (see section 5), and would thus tend to underestimate the overall importance of the iron cycle were we to select a reactivity constant that is too low. Finally, field evidence suggests that $FeOOH$ that is freshly precipitated is highly reactive (Picard et al., 2015; Beam et al., 2018) and thus iron precipitating from the surface ocean is expected to react on relatively short timescales. Any $FeOOH$ minerals that would resist reduction passing through a sulphidic water column are likely unreactive, and are thus presumably inert on early diagenetic timescales. Indeed, iron oxide minerals in sediments underlying euxinic water columns tend to show no depth trend, indicating that very little iron reduction is occurring at depth in such systems (see e.g., Xiong et al., 2019). Taken together, these observations support the presumption that once $Fe^{3+}$ is scavenged, its reactivity is less than hydrous ferric oxide but higher than that of goethite, motivating our default $k_{SMI,s}$ value. Nevertheless, the value used for this parameter should be considered carefully depending on the model application and assumed boundary conditions.

## 5    Outlook and conclusions

The principal aim of this paper is to provide a detailed description of our extension of the cGENIE biogeochemistry module to include coupled, anoxic Fe and S biogeochemical cycles. Because direct tuning with actual measured concentrations and

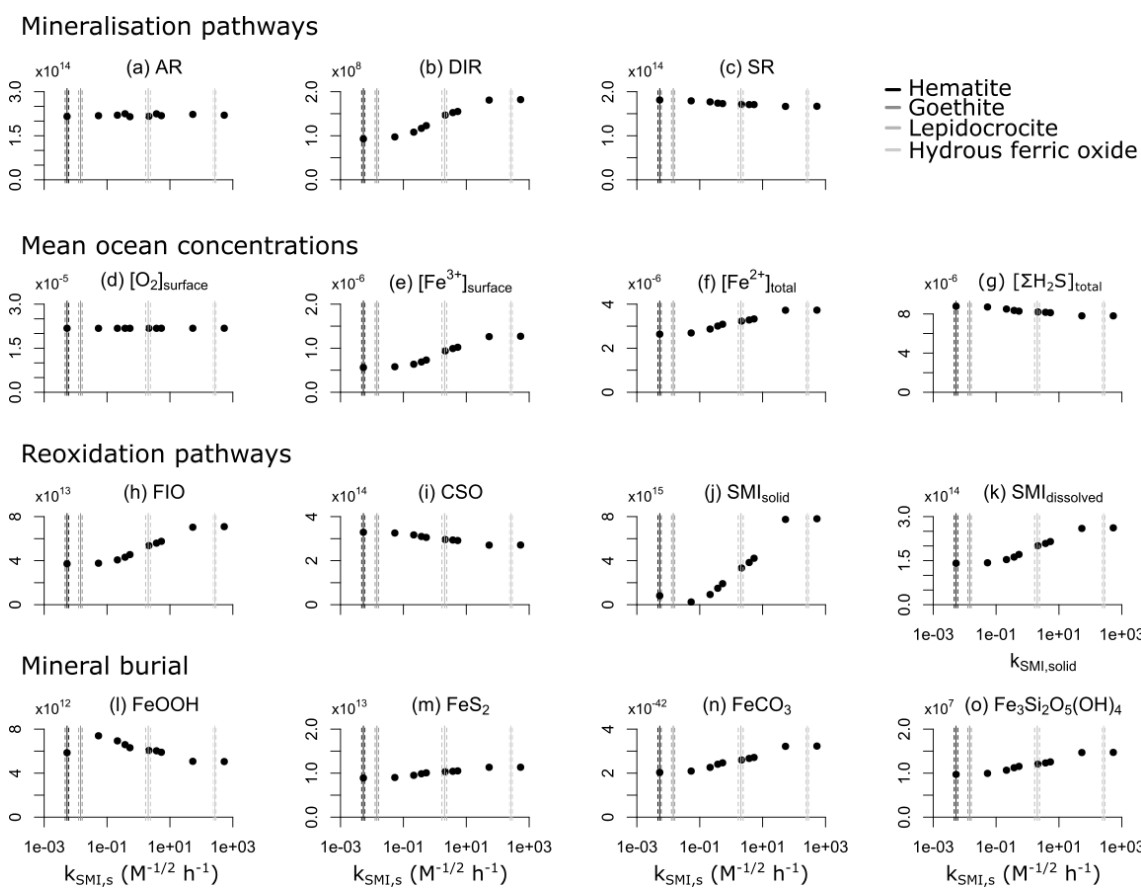

**Figure 13.** Global model output for a range of $k_{SMI,s}$ values (units are $M^{-1/2}\ h^{-1}$). Lines indicate experimental reactivity parameters as presented in (Poulton et al., 2004). Panels (a)-(c) and (h)-(n) are in $mol\ yr^{-1}$, panels (d)-(g) are in $mol\ kg^{-1}$. Note that the x-axis is logarithmic.

rates is not possible, we have relied heavily on kinetic constants and solubility values extracted from laboratory incubations. While care should be exercised in the application of these kinetic constants to reactions under *in-situ* conditions, our sensitivity analysis indicates that our key model results are robust across a wide range of possible parameter values. In addition, our proposed baseline parametrisation yields reaction rates, concentrations, burial fluxes, and stable Fe isotope compositions that

5    broadly compare well to both field measurements of process analogues for ancient ocean systems (i.e., anoxic lakes) and observations from the geologic record. Therefore, we believe that our model description of the anoxic Fe-S cycle is a valuable tool and an important step forward in simulating ocean redox landscapes during periods of Earth's history in which the ocean interior was pervasively anoxic. However, below we highlight some important limitations to our current model architecture, and also give some examples of possible future developments.

10    Most notably, our model is currently unable to resolve any sedimentary processes that would contribute to the global iron cycle (see e.g., Dale et al., 2015). In particular, the model does not include a representation of benthic iron reduction and

recycling back into the water column. Building on the improvements in the biogeochemistry of cGENIE described here, in the future we plan to extend the organic matter enabled sediment component of cGENIE (OMEN-SED; Hülse et al., 2018) to include an explicit representation of the benthic iron cycle. We anticipate that this will both improve the realism of tracer fields within the ocean interior and will make comparisons between predicted sedimentary signals and observations from Earth's

sedimentary rock record more accurate and robust. Second, there are likely to be important mechanistic links between the biogeochemistry of Fe and S within the ocean and the local and global recycling and bioavailability of key nutrient species for the biosphere (Bjerrum and Canfield, 2002; Laakso and Schrag, 2014; Jones et al., 2015; Reinhard et al., 2017). Future work will thus also focus on explicitly coupling the anoxic Fe and S biogeochemistry to the phosphorus (P) and nitrogen (N) cycles, and in particular the scavenging and remobilisation of P under different redox states and the impact of dissolved

Fe availability on nitrogen fixation. The modularity of cGENIE also allows the substitution of an explicit plankton ecological model ('ECOGEM') for the default biological export scheme (Ward et al., 2018), enabling the exploration of feedbacks between marine ecosystems, nutrient availability and ocean redox conditions (Reinhard et al., 2020b). Lastly, future work will seek to include other redox-sensitive proxies and bioessential elements, such as molybdenum, uranium or vanadium (Tribovillard, 2006) within the model code, which will further extend the applicability of our model and help to validate it against observations

from modern anoxic systems and the geologic record.

*Code availability.*  The code for the version of the 'muffin' release of the cGENIE Earth system model used in this paper, is tagged as v0.9.21, and is assigned a DOI: 10.5281/zenodo.4651390. The code is hosted on GitHub and can be obtained by cloning:

`https://github.com/derpycode/cgenie.muffin`

changing the directory to `cgenie.muffin` and then checking out the specific release:

`$ git checkout v0.9.21`

Configuration files for the specific experiments presented in the paper can be found in the directory: genie-userconfigs/MS/vandeveldeetal.GMD.2021

`genie-userconfigs/MS/vandeveldeetal.GMD.2021`

Details of the experiments, plus the command line needed to run each one, are given in the `readme.txt` file in that directory. All other configuration files and boundary conditions are provided as part of the code release. A manual detailing code installation, basic model

configuration, tutorials covering various aspects of model configuration and experimental design, plus results output and processing, is assigned a DOI: 10.5281/zenodo.4651394. The latex source of the manual, along with a pre-built PDF format version, can be obtained by cloning:

`https://github.com/derpycode/muffindoc`

**Table A1.** List of kinetic constants for the reactions included in the cGENIE model - converted to cGEnIE units.

| Reactivity constants | Symbol | Unit | Value |
|---|---|---|---|
| Limitation constant oxygen reduction | $K_{0,O_2}$ | $M$ | $8.0 \times 10^{-6}$ |
| Inhibition constant oxygen reduction | $K_{i,O_2}$ | $M$ | $8.0 \times 10^{-6}$ |
| Limitation constant DIR | $K_{0,FeOOH}$ | $M$ | $1.0 \times 10^{-2}$ |
| Inhibition constant DIR | $K_{i,FeOOH}$ | $M$ | $1.0 \times 10^{-2}$ |
| Limitation constant DSR | $K_{0,SO_4^{2-}}$ | $M$ | $5.0 \times 10^{-4}$ |
| Inhibition constant DSR | $K_{i,SO_4^{2-}}$ | $M$ | $1.0 \times 10^{-3}$ |
| Canonical sulphide oxidation | $k_{CSO}$ | $M^{-2}yr^{-1}$ | $5.5 \times 10^9$ |
| Ferrous iron oxidation | $k_{FIO}$ | $M^{-1}yr^{-1}$ | $1.0 \times 10^9$ |
| Sulphide-mediated iron reduction (dissolved) | $k_{SMI_d}$ | $M^{-0.5}yr^{-1}$ | $2.3 \times 10^6$ |
| Sulphide-mediated iron reduction (solid) | $k_{SMI_s}$ | $M^{-0.5}yr^{-1}$ | $1.7 \times 10^4$ |
| $Fe_{free}^{3+}$ Scavenging constant | $k_{scav}$ | $mol^{-1}m^2$ | $1.43 \times 10^{-6}$ |
| $L - Fe^{3+}$ complex stability constant | $K_{sp}^{FeL}$ | $M^{-1}$ | $1.0 \times 10^{11}$ |
| Solubility product $FeS_{aq}$ | $K_{sp}^{FeS_{aq}}$ | $M$ | $8.32 \times 10^{-6}$ |
| Kinetic constant pyrite precipitation | $k_{PyP}$ | $M^{-1}yr^{-1}$ | $3.25 \times 10^3$ |
| Kinetic constant siderite precipitation | $k_{AFC}$ | $Myr^{-1}$ | $1.72 \times 10^{-10}$ |
| Kinetic exponent siderite precipitation | $b_{AFC}$ | - | $9.042 \times 10^0$ |
| Solubility product greenalite | $K_{sp}^{greenalite}$ | $M^{-1}$ | $3.98 \times 10^{27}$ |
| Kinetic constant greenalite precipitation | $k_{greenalite}$ | $Myr^{-1}$ | $6.13 \times 10^{-9}$ |
| Kinetic exponent greenalite precipitation | $b_{greenalite}$ | - | $1.856 \times 10^0$ |
| *Activity coefficients* | | | |
| Activity constant $H^+$ | $\gamma_{H+}$ | - | 0.73 |
| Activity constant $OH^-$ | $\gamma_{OH^-}$ | - | 0.69 |
| Activity constant $CO_3^{2-}$ | $\gamma_{CO_3^{2-}}$ | - | 1.17 |
| Activity constant $Fe^{2+}$ | $\gamma_{Fe^{2+}}$ | - | 0.23 |
| Activity constant $SiO_2$ | $\gamma_{SiO_2}$ | - | 1.13 |

**Table A2.** Sensitivity range and baseline values of each parameter tested - converted to cGEnIE units.

| Constant | Symbol | Unit | Baseline | Minimum | Maximum |
|---|---|---|---|---|---|
| Limitation constant DIR | $K_{0,FeOOH}$ | $M$ | $1.0 \times 10^{-2}$ | $1.0 \times 10^{-5}$ | $1.0 \times 10^{1}$ |
| Inhibition constant DIR | $K_{i,FeOOH}$ | $M$ | $1.0 \times 10^{-2}$ | $1.0 \times 10^{-5}$ | $1.0 \times 10^{1}$ |
| Sulphide-mediated iron reduction (solid) | $k_{SMI,s}$ | $M^{-0.5}yr^{-1}$ | $1.73 \times 10^{-4}$ | $4.68 \times 10^{-1}$ | $4.68 \times 10^{6}$ |
| Pyrite precipitation | $k_{PyP}$ | $M^{-1}yr^{-1}$ | $3.25 \times 10^{3}$ | $3.25 \times 10^{0}$ | $3.25 \times 10^{5}$ |

# Appendix A

*Author contributions.* SJV and AR developed the model with input from all authors. CR provided geological context. SJV and DH ran the sensitivity tests and analysed the output. SJV wrote the paper with input from all authors.

*Competing interests.* The authors declare that they have no conflict of interest.

5   *Disclaimer.* TEXT

*Acknowledgements.* SVDV was supported by a post-doctoral Fellowship of the Belgian American Educational Foundation (2018-2019) and a NASA Postdoctoral Program fellowship at the University of California, Riverside, administered by Universities Space Research Association under contract with NASA (2019-2020). DH is supported by the Simons Foundation (Postdoctoral Fellowship in Marine Microbial Ecology, Award 653829). CR gratefully acknowledges funding from the NASA Exobiology Program and the NASA Nexus for Exoplanet System 10   Science (NExSS).

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
