# Peer review of "Iron and sulphur cycling in the cGENIE.muffin Earth system model (v0.9.21)"

_Geoscientific Model Development, 2020_

## Referee Comment (RC1) · Anonymous Referee #1 · 25 Nov 2020

van de Velde et al. titled "Anoxic iron and sulfur cycling in the cGENIE.muffin Earth system model (v0.9.16)" present an addition to the cGENIE model and corporate more substantial iron and sulfur cycles into the framework.

The authors highlight the importance and need of this work nicely and their manuscript offers a breakdown of the model that is clear and well written.

I support the ultimate publication of this manuscript, however I recommend the authors further explain the reasoning behind these decisions and expand upon the following points:

Why only anoxic cycling? The oxic cycle of these two elements is fundamental to modern day biogeochemical systems and likely systems that have areas that are oxic

but the deep ocean is predominantly anoxic. This then leads me to two questions. How has the model has been calibrated to recreate realistically expected changes in the iron and sulfur system. Their assumption that they can recreate anoxic analogues has been discussed in length in Chapter 4 however. Secondly, how does the model transition between oxic and anoxic iron and sulfur cycles, I can't seem to see that distinction? For example, in the early Phanerozoic and Neoproterozoic with regions of the shelf being oxic. Also if these new anoxic fluxes are included in a present-day Genie run, can it still recreate present day?

Why have you not attempted to couple the cycles together, surely simple tests can be included to see if the premise of justification of the model is correct? (i.e. a simple investigation of how the P cycle is effected by this new iron cycle?). I think a simple check is required as it is the reason this model has been built, based on your introduction.

Page10: The authors write "regardless of the relative availability of different electron acceptors, in any one depth interval in the ocean, exactly the same proportion of organic matter will be degraded". This means that regardless of the redox state of the ocean, the same amount of organic carbon will ultimately have to be produced and buried as to preserve the cascade of rigid degredation fluxes? Is this correct?

Page18 Lines13-17: I understand striving for simplicity however I believe the inclusion of a gypsum burial term is crucial to interpretations made from this model. As the formation and development of euxinia is traditionally dependant/associated with increases in sulfate input, it seems logical to suggest gypsum would become more quantitative post GOE (which is also seen within the rock record). Therefore, this assumption is only really adequate for pre-$\sim$2.5 Ga?

Page22 Lines 1-7: A 'closed' run of the model is a fair assumption but surely this is at odds with understanding the feedbacks of the Fe and S cycles on the overall systems dynamics? Likely restricting the interpretation and comparison to ancient analogues that you suggest is enough to test the model. For example, if feedbacks in the model

that will be introduced in future work meaning that the model cannot recreate the analogues then this is a problem.

Section 4.1: You initialise the model at a ratio of 2:1 for Fe and S. Which is fair under modern conditions but when considering the Precambrian Earth which you are here. The high S input flux to maintain this ratio with the elevated hydrothermal flux is presumably unreasonable based on the dominance of ferruginous conditions in the Precambrian. Surely the fluxes of non-sulfidized iron minerals should allow/produce a non- 2:1 burial of Fe:S during this interval?

Page22 Line 30: Why are Fe carbonates negligible? If you are recreating a largely anoxic ferruginous Precambrian ocean, surely a decent amount of $FeCO_3$ should precipitate? What is limiting it's precipitation? I understand the work by Tosca and co. but I struggle to see how it is restricted in this model as it is essentially based on iron and carbonate concentration? This could surely be an interesting finding, or simply a consequence of your high sulfate input? This issue leads the use of the model to compare to, for example iron speciation data, to not be adequate as many sediments in the Precambrian have Fe carbonates being a large proportion of the reactive iron. Hopefully the sedimentary model can correct this, but should be noted.

Page23: These fluxes relative to each other may well be a consequence of your elevated sulfate input relative to Fe and attempt to achieve a 2:1 ratio. This strict 2:1 ratio as far as I am aware is due to your exclusion of a gypsum burial flux, however a greater justification of this 2:1 ratio or testing different ratios is required.

Below are my minor corrections that I believe the manuscript will benefit from.

Page1 Line 4: I agree that these cycles impacted other elemental cycles such as phosphorus mentioned here. As the model is not fully integrated into the cGENIE model, would it be fair to add to the title ". . . in the cGENIE.muffin Earth system model framework. . ."?

Page1 Line 16: You say that you present patterns of concentrations of iron and it's isotopes, why not sulfur concentrations and isotopes? However in the abstract it is said that this work has been done, so why hasn't it been shown?

Page3 Line2: Poulton et al., 2005, should be Poulton and Canfield, 2005? – same for page4 Line29.

Throughout the text Poulton et al., 2004a is referenced. Your reference list only has one reference that fits Poulton et al., 2004, so the "a" needs to be removed from the citations.

Page6 Line 1-2: following Vervoort et al. (in review). This means that the wind and albedo methods are not currently published so I recommend it is added here for completeness.

Equations throughout: Are these yearly fluxes? Please clarify in the text. e.g. $\tau$bio=63.4 days. When this value is used in equation 1, is it input as 63.4 or (63.4/365) to convert to years? This issue is continued throughout. What resolution is the model running at, please clarify somewhere in text.

Page7 Line 26: Is this value of 0.66 fixed for all model runs? I would have thought this value would have varied at different points in geological time? Would be good to test this simply by varying between set values or adding a discussion on where this value comes from and why it is 0.66.

Page8: First time I read through this: I might have not understood it, in which case, it should be clarified, but is the export of organic carbon not related to the concentration of oxygen? As with the Precambrian you can have the same euphotic zone thickness but substantially different oxygen concentrations. After further reading however: This is clarified later, on page 9. Why is it worded like the above originally on page 8? All reads a bit contradictory. The method of remineralisation should be cleared up.

Page8 Line 12: "key interactions the key interactions" text repeated.

Page9 Equations: I don't see a difference between fDSR and fMG? Should there be? If so/if not, why?

Page11 Line 5: What is the concentration at which pyrite precipitation can occur? Also, in equation R9 of table 2, the pyrite precipitation doesn't seem to be dependent on iron concentrations and in the text, it says it is? Or at least, it is not discussed what the concentration of FeSp is until much later on, please define earlier to prevent hunting for the definition.

Page11 Line 10-15: Agreed that these four phases essentially make the iron redox proxy but greenalite is technically converted to magnetite or is at least heavily hypothesised to be. So, it should be clarified that it is not greenalite that is considered but magnetite.

Page12 Equation 11: Again, units seem inconsistent with the fluxes. Here, kscav is equal to mol-1m2h-1. Why are units now expressed in hours and not days or years? Go through full manuscript and ensure consistency or at least clarify what is put into the equation and what the flux units are.

Page12 Equation 12: What is Lfree? And as I am reading it, you seem to be doubling counting the Ltotal through this method as Feligand is bound to ligand? OR is Lfree, ligand that is not associated with Fe? Make this clear.

Page13 Line 8: "scavenged by POC"?

Page14 Line 25: A maximum rate of...?

Page16 Line 8: Independent of pH. This is a fair assumption for the current submission, however for further work it would be interesting to explore the impact of pH.

Page22 Lines 19-21: You say that these conditions are not meant to recreate any particular interval of Earth history but to justify initialising the model at 0.1PAL O2 and 50% PO4 input, you say this is done in order to simulate Precambrian conditions. I realise Archean vs. Proterozoic is significantly different, but you are clearly setting the

model for mid-Proterozoic conditions so why hide it in this sentence? Need to be more clear throughout the text, when in the Precambrian you are referring to.
* * *

---

## Referee Comment (RC2) · Anonymous Referee #2 · 8 Dec 2020

The modeling of pelagic Fe-S dynamics presented in the paper is a useful addition to the existing GENIE model framework. The reaction network developed by the authors considers most of the key processes involved in pelagic Fe-S cycling under anoxic conditions. I only found one model equation (Eq. 16, R7) that is probably not correct. In this equation, it is assumed that ferric iron oxidizes dissolved sulfide to sulfate while the evaluable data show that elemental sulfur is the major product of anaerobic sulfide oxidation with ferric iron oxides (e.g. Poulton et al. 2004). The elemental sulfur formed during this reaction is subsequently converted into pyrite by further reactions with sulfide and iron minerals under anoxic conditions. Since it would be difficult to introduce elemental sulfur as an additional tracer, I would recommend to use a more realistic stoichiometry for R7 where sulfide is converted into pyrite rather than sulfate. The au-

thors should at least show one additional model run with this revised stoichiometry to investigate to what extent the model output is affected by the artificial sulfate source introduced via R7.

The other problem of the paper is related to the lack of validation. The authors acknowledge that key processes such as the benthic iron cycle and the pelagic/benthic nitrogen cycle are not considered in their modeling framework. Since these processes play important roles, both in ancient oceans and modern lakes, it seems to be impossible to validate the model by field observations and proxy records. I do understand that the authors plan to expand their model to include these processes and that they see their manuscript as a step that needs to be taken to produce a more complete GENIE model version. To demonstrate the potential importance of their contribution, the authors should consider to embed their new pelagic Fe/S machinery into published box models that consider benthic Fe/S and nitrogen cycling to test how the performance of these models is enhanced by their pelagic Fe/S model equations.

The paper is well written and potentially useful. It could be further improved during the revision process by considering the comments above.

---

## Referee Comment (RC3) · Anonymous Referee #3 · 14 Dec 2020

General comments:

This paper is a nice clear introduction to the anoxic iron and sulfur cycling scheme which has been introduced into the cGENIE model. The limitations of the approach are well stated and the results and uncertainties are clearly displayed. I have a few minor comments below but overall the paper needs little modification in my opinion, given that it is very much a technical document.

Specific comments:

-page 5 line 25: a lot of the ideas the model is aimed towards involve biogeochemical cycling on the marine shelf and slope, but the model ocean is entirely abyssal. It would be worth noting this and giving some background here.

[Figure]

-Page 9. Why do you ignore nitrate reduction? What will you miss by ignoring this?

-Page 11: Table 2: the kinetic formulations are "based on standard kinetic formulations in biogeochemical models (see text for details)". Can you include the relevant references in the table alongside each expression? I am not sure I can see them all in the text.

-Figure 4: perhaps I am misreading this (?) but given that panel a has H2S at 100uM and panel b has Fe2 at 100um, then shouldn't the fractional values be the same for each plot when we look at 100uM on the x axis? E.g. Fe2(free) is 40% in panel a but only 20% in panel b under what appear to be the exact same conditions?

-Page 18 line 15: might note recent work that proposes an important role for gypsum in Precambrian oxygenation e.g. Shields et al (2019) nature geosci.

-Page 22: "Since the aim of this manuscript is to describe the newly developed Fe-S chemistry, and not ocean circulation in real or fake worlds, we will not further discuss the emerging patterns of ocean circulation." Yet they are discussed in the next section as they are fundamental for driving redox variability. There should be a brief summary of the circulation here.

Figure 9 b, should say "S mass balance" as the title

---

## Author Comment (AC1) · 6 Feb 2021

Title: "Anoxic iron and sulphur cycling in the cGENIE.muffin Earth system model (v0.9.16)"
Tracking #: gmd-2020-312
Authors: van de Velde et al.

Referee #1:

van de Velde et al. titled "Anoxic iron and sulfur cycling in the cGENIE.muffin Earth system model (v0.9.16)" present an addition to the cGENIE model and corporate more substantial iron and sulfur cycles into the framework. The authors highlight the importance and need of this work nicely and their manuscript offers a breakdown of the model that is clear and well written.

I support the ultimate publication of this manuscript, however I recommend the authors further explain the reasoning behind these decisions and expand upon the following points:

We appreciate the interest and the in-depth and constructive review effort. As we indicate below, we have addressed all of the concerns and expanded our reasoning in the manuscript.

Why only anoxic cycling? The oxic cycle of these two elements is fundamental to modern day biogeochemical systems and likely systems that have areas that are oxic but the deep ocean is predominantly anoxic. This then leads me to two questions.

Our description of the model developments seems to have created a misconception. The presented implementation of iron and sulphur cycling in the cGENIE.muffin model includes both oxic and anoxic cycling (as we discuss in Section 3.1 and 3.2). Oxic iron and sulphur cycling already existed in the previous version of the cGENIE.muffin model, and we extended the reaction set to include an anoxic cycle, in particulate iron reduction and formation of reduced Fe (and Fe-S) minerals. To avoid giving a false impression that only anoxic cycling of Fe and S is considered in the new version of the model, we made the following changes:

- the title is now '*Iron and sulphur cycling in the cGENIE.muffin Earth system model (v0.9.16)*' and

- the abstract at P1L11 now reads: '*Here, we extend the 'muffin' release of the intermediate-complexity Earth system model cGENIE, to now include an anoxic iron and sulphur cycle (expanding the existing oxic iron and sulphur cycles), ...*'

- the introduction at P4L23 now reads: '*In this paper, we present the development of a coupled anoxic oceanic iron and sulphur cycle, embedded within the 'muffin' release of the carbon-centric Grid ENabled Integrated Earth system model, 'cGENIE' (note that the oxic cycle of both Fe and S already exists in previous versions; Tagliabue et al., 2016).*'

How has the model has been calibrated to recreate realistically expected changes in the iron and sulfur system. Their assumption that they can recreate anoxic analogues has been discussed in length in Chapter 4 however.

Our model developments consisted of implementing a set of well-established mechanistic biogeochemical reactions, using mostly well-constrained kinetic parameters. The oxic part of the Fe cycle has been calibrated against modern observations – as discussed in Tagliabue et al. (2016). For the novel reaction set, as we discuss in the text, the baseline parametrisation of the

reactions included in the Fe-S system all rely on laboratory measurements of the kinetic rates of these respective reactions. Because we did not have to parametrise 'black-box' reactions, there was no real need for extensive calibration of the model, which is supported by the fact that the baseline parametrisation is able to simulate realistic rates and concentrations, when compared to other anoxic systems, as we show in Chapter 4 (as also indicated by the reviewer).

We have added this rationale to the text at P22L12: '*As a result, we lack observations to which we can directly calibrate our model. However, as our model development consisted of the implementation of well-established, mechanistic biogeochemical reactions, with relatively well-defined kinetic rates, our model should simulate realistic rates and concentrations without extensive calibration of model parameters. We illustrate this by showing the spatial concentration and isotope features of a hypothetical anoxic ocean (section 4.2), and -- where possible -- compare our predicted reaction rates to rates obtained from anoxic process analogues for the ancient oceans.*'

Secondly, how does the model transition between oxic and anoxic iron and sulfur cycles, I can't seem to see that distinction? For example, in the early Phanerozoic and Neoproterozoic with regions of the shelf being oxic.

It is absolutely correct that the spatial ocean redox structure has evolved in complex ways throughout Earth's history, and has potentially contained regions that were oxygenated while other parts contained ferrous iron or dissolved sulphide. Indeed, the explicit aim of our model development was to be able to simulate these spatial redox patterns. As we show in Figure 7 (P26), with the new reaction set, cGENIE.muffin is able to recreate spatial redox patterns as mentioned by the reviewer. Transitions between oxic and anoxic Fe/S cycling are not defined *a priori*, but are an emergent property of the model set-up. Simply put, once oxygen is consumed within a grid-cell, anoxic cycles will become dominant. As long as oxygen is available in a grid cell, reduced species of Fe and S will be oxidised, and mineralisation of organic matter will be via aerobic respiration. Once oxygen becomes exhausted, organic matter mineralisation will be coupled to iron and sulphate reduction, creating reduced species of Fe and S. Ocean transport will bring these reduced species (which will build up in the ocean interior) to the surface, where they will react with oxygen (via upwelling), and will also bring oxygen to the ocean interior (via downwelling).

To clarify the mechanisms regulating the transition between the oxic and anoxic cycles we have adapted Figure 4b (P9), so it shows there is a connection between both parts driven by transport of oxidised/reduced species (indicated by the dashed lines).

Also if these new anoxic fluxes are included in a present-day Genie run, can it still recreate present day?

That is a good point, and we set-up a modern configuration comparing the previous cGEnIE configuration (without an anoxic iron cycle) and the new developed cGEnIE configuration (including an anoxic iron cycle) to illustrate that there is very little difference between the performance of the model under fully oxygenated conditions. The figure below compares total dissolved Fe ([TDFe] = [Fe$^{3+}$]+[Fe$^{2+}$], not that Fe$^{2+}$ only exists in the new scheme) for both the old scheme and our extension. Our extension shows slightly higher total dissolved iron just below the surface around the equator – which is due to an additional release of Fe$^{2+}$ in the Oxygen-Minimum-Zones, which does not occur in the old redox-insensitive scheme.

We will include a similar analysis and discussion in the revised paper.

[Figure]

Why have you not attempted to couple the cycles together, surely simple tests can be included to see if the premise of justification of the model is correct? (i.e. a simple investigation of how the P cycle is effected by this new iron cycle?). I think a simple check is required as it is the reason this model has been built, based on your introduction.

The coupling to the P cycle is most certainly not the only reason we have built this model - we apologize for lacking clarity on this point. The usefulness of simulating spatial redox patterns will go well beyond questions regarding P-limitation in anoxic oceans. Our introduction highlights several questions that could also benefit from a spatially explicit Fe-S cycle, not least with regard to aiding in the interpretation of redox proxies from the geological record (P3L7-11), but also in addressing questions of anoxygenic ecosystems before the advent of oxygenic photosynthesis (P2L19-24).

We respectfully disagree with the reviewer that coupling the Fe and P cycles constitutes a simple check. This is an entirely different code development effort, targeted at a distinct set of questions. We consider it a crucial first step to develop the underlying framework for this effort by including both oxic and anoxic Fe recycling within the 3-D biogeochemical framework in GENIE.muffin, and then in a subsequent paper (currently in prep), exhaustively characterise and focus on the role of scavenging interactions between Fe and P and associated nutrient feedbacks.

Page10: The authors write "regardless of the relative availability of different electron acceptors, in any one depth interval in the ocean, exactly the same proportion of organic matter will be degraded". This means that regardless of the redox state of the ocean, the same amount of organic carbon will ultimately have to be produced and buried as to preserve the cascade of rigid degredation fluxes? Is this correct?

This is partially correct – the production of organic matter is controlled by the availability of dissolved nutrients (in the case of this manuscript: $PO_4^{3-}$ and $Fe^{3+}$), and is irrespective of the ocean redox state. The degradation of organic carbon is controlled by the (i) production of organic carbon in the photic zone and (ii) the shape of the 'Martin'-curve as shown in Fig. 4a (P9). It is not the fluxes that are rigid, but the fraction of degradation (see for example in Fig. 4a, at 1 km depth, less than 20% of the organic matter initially produced remains).

We now clarify this at P9L16: '*A 'Martin'-type decay curve of organic matter flux with depth is prescribed in the model, such that regardless of the relative availability of different electron acceptors, the fraction of organic matter that will be degraded (relative to the amount of organic matter produced in the photic zone)is depth-dependent.*'

Page18 Lines13-17: I understand striving for simplicity however I believe the inclusion of a gypsum burial term is crucial to interpretations made from this model. As the formation and development of euxinia is traditionally dependant/associated with increases in sulfate input, it seems logical to suggest gypsum would become more quantitative post GOE (which is also seen within the rock record). Therefore, this assumption is only really adequate for pre-2.5 Ga?

The reviewer raises an interesting point, and we did not mean to suggest that gypsum did not precipitate prior to the Phanerozoic. Nevertheless, global compilations, such as the one presented in Canfield and Farquhar (2009), indicate that, from a global mass balance perspective, most sulphur left the ocean in the form of pyrite (this is also suggested for the Phanerozoic; Halevy et al., 2012). This does not exclude however that gypsum precipitate locally, but several considerations prevent the straight-forward implementation of gypsum precipitation. First, gypsum is an evaporite mineral, and thus predominantly precipitates in location where seawater evaporates, leading to supersaturation of gypsum (also halite) and consequent precipitation of these minerals. These processes are rather episodic and occur under very specific (regional) conditions that are not easily simulated with the GENIE.muffin owing to its low resolution especially for shelf environment. Secondly, if we were to include gypsum as a sulphate mineral, we would also have to make additional assumptions with respect to $Ca^{2+}$ concentrations, which have been likely variable in the past, but actual concentrations are uncertain. Nevertheless, we aim to include gypsum burial in future developments of the GENIE.muffin code, to allow the simulation of transient events where sulphate delivery to the ocean is temporarily enhanced.

We have extended our reasoning for omitting gypsum burial at this instance at P19L21: '*We do not currently include precipitation of gypsum CaSO4 in our model description (Fig. 4b). Gypsum is an evaporite mineral that precipitates during regional and episodic events of supersaturation, and was likely a less important sulphur sink on a globally integrated basis during Precambrian time, or during any other period in which ocean SO42- was relatively low (Grotzinger et al., 1993, Crowe et al., 2014, Fakhraee et al., 2019). Indeed, there is still some debate as to the time-integrated impact of sulfate evaporites on the steady-state global sulfur cycle even during more recent periods of Earth's history (Halevy et al., 2012; Canfield, 2013). However, due to its episodic nature, gypsum could play an important role as a sulphate source during transient events (for example during events of enhanced weathering of a gypsum-rich source; Shields et al., 2019). A planned future development to the cGENIE.muffin model is the incorporation of an explicit gypsum cycle, which will allow the investigation of transient events of enhanced sulphate delivery to the ocean (see, e.g., Shields et al., 2019).*'

Page22 Lines 1-7: A 'closed' run of the model is a fair assumption but surely this is at odds with understanding the feedbacks of the Fe and S cycles on the overall systems dynamics? Likely restricting the interpretation and comparison to ancient analogues that you suggest is enough to test the model. For example, if feedbacks in the model that will be introduced in future work meaning that the model cannot recreate the analogues then this is a problem.

The reviewer raises a fair point. In the current version of the model, however, the Fe/S cycles are dependent on the primary productivity and chosen boundary conditions (e.g., atmospheric oxygen concentrations, hydrothermal Fe fluxes …). While in reality there will be feedbacks between ocean redox and productivity (and ultimately atmospheric oxygen concentrations), these are not yet included in our current model development. Our ultimate goal is to include these feedbacks, and investigate their relative strengths or importance. However, extending the model complexity and reaction set will require time, and we believe that the current model development presents enough advancement to be published in its current form. By considering a closed system for carbon and phosphorus, we can fix important boundary conditions ($pCO_2$, ocean productivity, ocean circulation …) and investigate the primary controls on ocean redox state in a spatially explicit and well-controlled way. Indeed, there is a range of applications for which such an approach will be useful, more transparent, and easier to understand than modelling the overall system.

We give more reasoning for the choice of a closed configuration for C and P at P23L14: '*The model is run in 'closed' configuration for all elements (notably C and P), in which the ocean-atmosphere inventory for each element is always conserved. We chose a closed configuration for C and P, because it allows us to fix important boundary conditions (such as $pCO_2$ and productivity), and quantify the emerging, steady-state ocean redox state under these conditions.*'

Section 4.1: You initialise the model at a ratio of 2:1 for Fe and S. Which is fair under modern conditions but when considering the Precambrian Earth which you are here. The high S input flux to maintain this ratio with the elevated hydrothermal flux is presumably unreasonable based on the dominance of ferruginous conditions in the Precambrian. Surely the fluxes of non-sulfidized iron minerals should allow/produce a non- 2:1 burial of Fe:S during this interval?

The reviewer is absolutely correct, and we are also aware of that (see P23L27). We chose this particular model set-up as an example of model output, without choosing a specific Precambrian time period, and this flux ratio was chosen because we did not want to introduce a bias towards any dominating redox state (P23L29). The burial fluxes do indeed generate a different Fe:S ratio (1.3:1 to be exact; see Table 5).

Page22 Line 30: Why are Fe carbonates negligible? If you are recreating a largely anoxic ferruginous Precambrian ocean, surely a decent amount of FeCO3 should precipitate? What is limiting it's precipitation? I understand the work by Tosca and co. but I struggle to see how it is restricted in this model as it is essentially based on iron and carbonate concentration? This could surely be an interesting finding, or simply a consequence of your high sulfate input? This issue leads the use of the model to compare to, for example iron speciation data, to not be adequate as many sediments in the Precambrian have Fe carbonates being a large proportion of the reactive iron. Hopefully the sedimentary model can correct this, but should be noted.

Fe carbonates are a negligible burial flux of Fe from the anoxic ocean interior because their formation requires extreme degrees of supersaturation (Jimenez-Lopez and Romanek, 2004; Dideriksen et al., 2015; Jiang and Tosca, 2019). These levels of Fe carbonate saturation are not reached in our model under the examined boundary conditions. As noted by these authors, and as corroborated by our baseline simulation, other Fe-species (such as greenalite) will precipitate before siderite precipitation becomes important. In our case, Fe-concentrations seem to be controlled by the formation of FeOOH minerals in the oxic parts of the ocean (see Table 5). Our results seem to indicate that the formation of Fe(II) minerals in the sediment are critical to allow comparison between our model output and the iron speciation proxies, and the development of a coupled sedimentary model is the focus on ongoing work.

We added more explanation to the observation of negligible $FeCO_3$ fluxes at P24L27: '*Fluxes of $FeCO_3$ where near-zero everywhere (data not shown), consistent with recent work suggesting that water column precipitation of $FeCO_3$ may be difficult to achieve, even in iron-dominated oceans (Jiang and Tosca 2019; Tosca et al., 2019). The negligible $FeCO_3$ fluxes are at odds with Fe-speciation data of Precambrian rocks that show an important fraction of sedimentary Fe consists of reduced non-sulphurised Fe minerals (Sperling et al., 2015). Our results indicate that these minerals are most likely formed during sedimentary diagenesis, emphasizing the potential importance of processes below the sediment-water interface in structuring Fe-speciation signals in Earth's rock record. Future development will thus include a representation of sedimentary Fe cycling in the sedimentary module 'OMEN-SED' (Hülse et al., 2018).*'

Page23: These fluxes relative to each other may well be a consequence of your elevated sulfate input relative to Fe and attempt to achieve a 2:1 ratio. This strict 2:1 ratio as far as I am aware is due to your exclusion of a gypsum burial flux, however a greater justification of this 2:1 ratio or testing different ratios is required.

The burial fluxes of individual Fe-compounds are indeed a consequence of the model set-up and assumptions made about (relative) fluxes of Fe and S. We did not attempt to achieve a 2:1 ratio in burial flux, but rather chose to deliver Fe and S in a 2:1 ratio to, as explained above, avoid bias in the ocean redox state. Because we are not attempting to reproduce any particular time-period, which we emphasise at P23L31, we believe it is not possible or necessary to provide more justification than the one given in the main text for the chosen boundary conditions. This manuscript is meant as a basic description of the new model code developments, and basic demonstration of the robustness of the new model parameters. Potential future model studies that aim to tackle specific periods in Earth's history, which we fully agree with the reviewer would require more involved justification of the chosen boundary conditions.

Below are my minor corrections that I believe the manuscript will benefit from.

Page1 Line 4: I agree that these cycles impacted other elemental cycles such as phosphorus mentioned here. As the model is not fully integrated into the cGENIE model, would it be fair to add to the title ": : in the cGENIE.muffin Earth system model framework: : :"?

The new model equations are fully integrated into the biogeochemical module 'BIOGEM' of the cGEnIE.muffin model. So we feel that it would be more correct to keep the title as it is.

Page1 Line 16: You say that you present patterns of concentrations of iron and it's isotopes, why not sulfur concentrations and isotopes? However in the abstract it is said that this work has been done, so why hasn't it been shown?

That was a mistake from our side, as we do present S-isotope patterns. This has been corrected in the abstract.

Page3 Line2: Poulton et al., 2005, should be Poulton and Canfield, 2005? – same for page4 Line29.

This has been corrected

Throughout the text Poulton et al., 2004a is referenced. Your reference list only has one reference that fits Poulton et al., 2004, so the "a" needs to be removed from the citations.

This has been corrected

Page6 Line 1-2: following Vervoort et al. (in review). This means that the wind and albedo methods are not currently published so I recommend it is added here for completeness.

We have added albedo and wind stress on the previous Fig. 5 and moved it forward (so it is now Fig. 3)

Equations throughout: Are these yearly fluxes? Please clarify in the text. e.g. bio=63.4 days. When this value is used in equation 1, is it input as 63.4 or (63.4/365) to convert to years? This issue is continued throughout. What resolution is the model running at, please clarify somewhere in text.

We have checked our units to make them consistent, and clarify where the units were not given. For the equations, all rates are expressed in M h$^{-1}$, and fluxes as mol m$^{-2}$ h$^{-1}$. For the discussion however, we have converted the output yearly fluxes or reaction rates, as these are more directly comparable to geochemical literature. The model is running at 48 timesteps per year, which we have now clarified in the text at P5L15. '*the default temporal resolution, which we use here, is 48 timesteps per year*'

Page7 Line 26: Is this value of 0.66 fixed for all model runs? I would have thought this value would have varied at different points in geological time? Would be good to test this simply by varying between set values or adding a discussion on where this value comes from and why it is 0.66.

This value is fixed for all model runs, and has been assigned following the assumptions of the OCMIP-2 protocol (Najjar and Orr, 1999), we have added this citation in the main text: '*The value of v has been assigned following the assumptions of the OCMIP-2 protocol (Najjar and Orr, 1999; Ridgwell et al., 2007)*'. Although it is certainly possible that this parameter has changed through time, there are no empirical constraints on this. As a result, we have left a full exploration of the sensitivity to this parameter for future work, and opt to utilize the well-established value for ease of comparison with previous work.

However, in a recent publication (Crichton, K. A., J. D. Wilson, A. Ridgwell, P. N. Pearson, Calibration of temperature-dependent ocean microbial processes in the cGENIE.muffin (v0.9.13) Earth system model, GMD, 10.5194/gmd-14-125-2021 (2021)) we did implement and test a new option for enacting temperature-dependent partitioning (and remineralization) of

DOM, and we will highlight this alternative configuration possibility in the revision. P8L5 '*Recently, a new option for enacting temperature-dependent partitioning (and remineralisation) of DOP has been implemented (Crichton et al., 2021), which presents an alternative to the fixed DOP/POP partitioning used here.*'

Page8: First time I read through this: I might have not understood it, in which case, it should be clarified, but is the export of organic carbon not related to the concentration of oxygen? As with the Precambrian you can have the same euphotic zone thickness but substantially different oxygen concentrations. After further reading however: This is clarified later, on page 9. Why is it worded like the above originally on page 8? All reads a bit contradictory. The method of remineralisation should be cleared up.

The export flux is indeed independent of the concentration of oxygen in GENIE.muffin. We assume here that POC mineralisation rate is driven by the magnitude of the POC flux, which is prescribed as following a Martin-curve. We have moved the clarification forward, to avoid the initial confusion: '*Redox cycling in the ocean (and sediments) is driven by the mineralisation of POC, which is produced in the photic zone by photosynthesis, and subsequently sinks through the water column (Ridgwell et al., 2007). A 'Martin'-type decay curve of organic matter flux with depth is prescribed in the model, such that regardless of the relative availability of different electron acceptors, the fraction of organic matter that will be degraded (relative to the amount of organic matter produced in the photic zone)is depth-dependent. We avoid the alternative here -- a fully kinetic set of equations where each electron acceptor is associated with a different rate of degradation -- partly because of the additional set of poorly constrained (kinetic rate constant) parameters that would be required, and partly because to implement such a scheme effectively, requires knowledge about the composition of settling organic matter and how its relative reactivity changes with time (Ridgwell, 2011; Larowe and Van Cappellen, 2011). Thus, the mineralisation rate is dependent on the magnitude of the POC flux, which follows a 'Martin'-type decay (Fig. 3a), which determines the mineralisation rate (Rmin) at each depth layer in the water column*'*

Page8 Line 12: "key interactions the key interactions" text repeated.

Corrected

Page9 Equations: I don't see a difference between fDSR and fMG? Should there be? If so/if not, why?

There is a difference between $f_{DSR}$ and $f_{MG}$. For fDSR $[SO_4^{2-}]$ is in the nominator, whereas for $f_{MG}$ it is $K_{i,SO4}$. These expressions are conventional inhibition-limitation equations. At high $SO_4^{2-}$ concentrations (higher than $K_{i,SO4}$) methanogenesis is inhibited. When $SO_4^{2-}$ concentrations drop, sulphate reduction becomes limited and methanogenesis becomes important.

Page11 Line 5: What is the concentration at which pyrite precipitation can occur? Also, in equation R9 of table 2, the pyrite precipitation doesn't seem to be dependent on iron concentrations and in the text, it says it is? Or at least, it is not discussed what the concentration of FeSp is until much later on, please define earlier to prevent hunting for the definition.

We described the pyrite formation process in more detail at P11L5: '*When Fe2+ and H2S are simultaneously present in the water column they form dissolved FeS (FeSaq). Once FeSaq*

*surpasses a solubility threshold of ~ 2 μM (Rickard, 2006), particulate FeSp can form, which further reacts with H2S to form the mineral pyrite (FeS2) via pyrite precipitation*'

Page11 Line 10-15: Agreed that these four phases essentially make the iron redox proxy but greenalite is technically converted to magnetite or is at least heavily hypothesised to be. So, it should be clarified that it is not greenalite that is considered but magnetite.

That is a good point, we have added clarification at P11L15: '*It should be noted however that some of these phases can undergo transformations to other phases in the sediment after deposition (such as greenalite to magnetite). In our current model set-up, no sedimentary processes are included, but future developments will address the sedimentary part of the Fe and S cycle.*'

Page12 Equation 11: Again, units seem inconsistent with the fluxes. Here, kscav is equal to mol-1m2h-1. Why are units now expressed in hours and not days or years? Go through full manuscript and ensure consistency or at least clarify what is put into the equation and what the flux units are.

See our answer to the point raised above. The parameter $k_{scav}$ should be in $mol^{-1}$ $m^2$, so the units of the Eq. 11 become $M$ $h^{-1}$. This has now been corrected.

Page12 Equation 12: What is Lfree? And as I am reading it, you seem to be doubling counting the Ltotal through this method as Feligand is bound to ligand? OR is Lfree, ligand that is not associated with Fe? Make this clear.

Lfree is ligand unassociated with Fe3+, this has now been clarified: '*Lfree is ligand unassociated with Fe3+ and Ltotal is the total amount of ligand*'

Page13 Line 8: "scavenged by POC"?

Corrected

Page14 Line 25: A maximum rate of: : :?

The maximum rate set by $R_{min}$. This has been clarified: '(the maximum rate is set by Rmin; see Eq. 10 and Table 2)'

Page16 Line 8: Independent of pH. This is a fair assumption for the current submission, however for further work it would be interesting to explore the impact of pH.

That is a good suggestion.

Page22 Lines 19-21: You say that these conditions are not meant to recreate any particular interval of Earth history but to justify initialising the model at 0.1PAL O2 and 50% PO4 input, you say this is done in order to simulate Precambrian conditions. I realise Archean vs. Proterozoic is significantly different, but you are clearly setting the model for mid-Proterozoic conditions so why hide it in this sentence? Need to be more clear throughout the text, when in the Precambrian you are referring to.

Our aim was to select boundary conditions that would generate an ocean that would exhibit the major different redox states (oxic, sulphidic and ferruginous). We opted to set up a baseline run with boundary conditions that broadly represent Proterozoic conditions, but these are not necessarily meant to be representative of any specific time period. Simulating particular periods

of Earth's history, with proper attention paid to specific boundary conditions, is an important task for future work (see also above).

Referee #2:

The modeling of pelagic Fe-S dynamics presented in the paper is a useful addition to the existing GENIE model framework. The reaction network developed by the authors considers most of the key processes involved in pelagic Fe-S cycling under anoxic conditions.

I only found one model equation (Eq. 16, R7) that is probably not correct. In this equation, it is assumed that ferric iron oxidizes dissolved sulfide to sulfate while the evaluable data show that elemental sulfur is the major product of anaerobic sulfide oxidation with ferric iron oxides (e.g. Poulton et al. 2004). The elemental sulfur formed during this reaction is subsequently converted into pyrite by further reactions with sulfide and iron minerals under anoxic conditions. Since it would be difficult to introduce elemental sulfur as an additional tracer, I would recommend to use a more realistic stoichiometry for R7 where sulfide is converted into pyrite rather than sulfate. The authors should at least show one additional model run with this revised stoichiometry to investigate to what extent the model output is affected by the artificial sulfate source introduced via R7.

The reviewer is correct that elemental sulphur is the product of sulphide oxidation coupled to iron oxide reduction. However, elemental sulphur can not only react with FeS, but is metastable and will also disproportionate to $SO_4^{2-}$ and $H_2S$ (see, e.g., Finster et al., 1998). Combining both reactions then yields the same stoichiometric equation as we give in the manuscript

$$\left| \begin{aligned} H_2S + 2FeOOH + 4H^+ &\rightarrow S^0 + 2Fe^{2+} + 4H_2O \\ S^0 + H_2O &\rightarrow \tfrac{3}{4}H_2S + \tfrac{1}{4}SO_4^{2-} + \tfrac{1}{2}H^+ \end{aligned} \right.$$
$$\overline{\tfrac{1}{4}H_2S + 2FeOOH + \tfrac{7}{2}H^+ \rightarrow \tfrac{1}{4}SO_4^{2-} + 2Fe^{2+} + 3H_2O}$$

Using the reaction combination suggested by the reviewer; sulphide oxidation coupled to direct precipitation of pyrite, i.e.:

$$\left| \begin{aligned} H_2S + 2FeOOH + 4H^+ &\rightarrow S^0 + 2Fe^{2+} + 4H_2O \\ Fe^{2+} + H_2S + S^0 &\rightarrow FeS_2 + 2H^+ \end{aligned} \right.$$
$$\overline{2H_2S + 2FeOOH + 2H^+ \rightarrow FeS_2 + Fe^{2+} + 4H_2O}$$

Forces pyrite to precipitate at a rate that is determined by the reduction of iron oxide with sulphide. It also implies that the rate of pyrite precipitation is dependent on the concentration of $H_2S$, but independent of the availability of $Fe^{2+}$ (or FeS). This is at odds with both known pathways of pyrite precipitation (the 'sulphide pathway' and 'elemental sulphur pathway'), both of which require FeS, and the rates of which are dependent on the concentration of FeS (Rickard, 1975):

$$FeS + S^0 \rightarrow FeS_2$$
$$FeS + H_2S \rightarrow FeS_2 + H_2$$

More recent work (Wan et al., 2017) has also shown that the precipitation of pyrite during the reduction of Fe(III) with sulphide is slow in a situation where $H_2S$ is in excess compared to Fe(III). Under these circumstances, the rate of pyrite precipitation is controlled by the 'sulphide-pathway', which is the one included in our model (R9 in Table 2). It is only in the inverse

situation (Fe(III) in excess compared to $H_2S$) that the reaction pathway suggested by the reviewer is dominant. However, even then the kinetics of pyrite formation are relatively slow (on the order of days) compared to the kinetics of the $H_2S$ oxidation. This situation is unlikely to be found in the water column, as Fe(III) will only persist in oxygenated waters (that do not contain $H_2S$), or as sinking particulates through the water column. The rapid settling of oxidised, particulate Fe(III) through the water column would prevent high concentrations of Fe(III). To achieve an excess of Fe(III) over $H_2S$, the $H_2S$ concentrations would have to be even lower, leading to very negligible rates of iron reduction. This reaction pathway may become important in the sediment of low-sulphate ocean (i.e. Archean-Proterozoic).

Nevertheless, we re-ran our baseline experiment with the alternative reaction pathway as suggested by the reviewer (see figures below), and as the figures illustrate, there is very little effect of changing the reaction pathway.

[Figure]

We have also revised the text to explicitly discuss our reasoning behind the stoichiometry of Eq. 16, at P15L25; '*Oxidised iron in the ocean can also be reduced via sulphur-mediated iron reduction (SMI), which follows the stoichiometry (Poulton et al., 2004):*

*$H_2S + 2Fe^{3+} \rightarrow S^0 + 2Fe^{2+} + 2H^+$  (16)*

*We are not explicitly modelling elemental sulphur ($S^0$), but assume that it becomes quantitatively disproportionated into $H_2S$ and $SO_4^{2-}$ (Finster et al., 1998),*

*$S^0 + H_2O \rightarrow \frac{3}{4} H_2S + \frac{1}{4} SO_4^{2-} + \frac{1}{2} H^+$ (17)*

*which then leads to the overall stoichiometry*

*$H_2S + 8Fe^{3+} + 4H_2O \rightarrow SO_4^{2-} + 8Fe^{2+} + 10H^+$  (18)*

*Note that [$Fe^{3+}$] in Eq. 19 can represent dissolved $Fe^{3+}$total or solid FeOOH. The assumption of quantitative disproportionation implies that pyrite precipitation is not closely coupled to the reaction between $H_2S$ and $Fe^{3+}$, but only occurs via precipitation of FeSp with $H_2S$ (see Section 3.1.3). Laboratory experiments have shown that this assumption is valid for aquatic systems where $Fe^{3+}$ is not in excess with respect to $H_2S$ (Wan et al., 2017), which is the case for most modern marine systems. We contend that this is also a valid assumption for water-column chemistry for most of Earth's history, as rapid settling of oxidised, particulate FeOOH through the water column would prevent high concentrations of $Fe^{3+}$ (in the water column). To achieve an excess of $Fe^{3+}$ over $H_2S$, the $H_2S$ concentrations would have to be even lower, leading to very negligible rates of iron reduction. However, this reaction pathway could become important in the sediment of low-sulphate ocean (i.e. periods of Archean time).*'

The other problem of the paper is related to the lack of validation. The authors acknowledge that key processes such as the benthic iron cycle and the pelagic/benthic nitrogen cycle are not considered in their modeling framework. Since these processes play important roles, both in ancient oceans and modern lakes, it seems to be impossible to validate the model by field observations and proxy records. I do understand that the authors plan to expand their model to include these processes and that they see their manuscript as a step that needs to be taken to produce a more complete GENIE model version. To demonstrate the potential importance of their contribution, the authors should consider to embed their new pelagic Fe/S machinery into published box models that consider benthic Fe/S and nitrogen cycling to test how the performance of these models is enhanced by their pelagic Fe/S model equations.

It is indeed not trivial to validate our model results, given that the current ocean is almost completely oxygenated. As we mentioned in our reply to Reviewer 1, our model developments consisted of implementing a set of well-established mechanistic biogeochemical reactions, using mostly well-constrained kinetic parameters. Where possible, we have compared our model output to available data from anoxic process analogues of ancient oceans (principally anoxic lake systems), and show that concentrations and reaction rates fall close to measured rates from natural analogue environments. In addition, we demonstrate that our primary model results are insensitive with respect to most parameter choices, indicating that the model behaviour is robust. We are not sure which published box models the reviewer is referring to, but it is no trivial task to develop models (even box models). Furthermore, the issue of validation will also exist for these box-models, and so we are not sure what would be learned from implementing this reaction set in a box-model.

The paper is well written and potentially useful. It could be further improved during the revision process by considering the comments above.

We are thankful for the positive evaluation of our manuscript.

Referee #3:

General comments:

This paper is a nice clear introduction to the anoxic iron and sulfur cycling scheme which has been introduced into the cGENIE model. The limitations of the approach are well stated and the results and uncertainties are clearly displayed. I have a few minor comments below but overall the paper needs little modification in my opinion, given that it is very much a technical document.

We are thankful for the positive evaluation of our manuscript.

Specific comments:

-page 5 line 25: a lot of the ideas the model is aimed towards involve biogeochemical cycling on the marine shelf and slope, but the model ocean is entirely abyssal. It would be worth noting this and giving some background here.

That is a correct remark. We have added more background for our choice of continental configuration at P6L2: '*It is worth noting that our model setup is entirely abyssal, while in reality the continental slopes and shelves are of greater importance for the biogeochemical cycling of Fe and S. We have chosen our idealized bathymetry and continental configuration as it generates a relatively simple ocean circulation and thus facilitates the interpretation of model output (see below). Our choice allows us to clearly illustrate the dependence of model output on the choice of parameters, whereas more elaborate continental configurations could introduce more complexity and obscure the model sensitivity to parameter selection.*'

-Page 9. Why do you ignore nitrate reduction? What will you miss by ignoring this?

We ignore nitrate reduction because the nitrogen cycle in cGEnIE is currently under development (as an extension to that described in Naafs, B.D.A., F.M. Monteiro, A. Pearson, M.B. Higgins, R.D. Pancost, and A. Ridgwell, Fundamentally different global marine nitrogen cycling in response to severe ocean deoxygenation, PNAS, DOI: 10.1073/pnas.1905553116 (2019)). For our development manuscript, the omission of a nitrogen cycle should not significantly affect the model output as we focus on evaluating simulated Fe and S dynamics and are not interested in for instance quantifying/comparing the different degradation pathways. But the reviewer is correct and for more targeted studies into nutrient limitation, a nitrogen cycle would need to be included.

-Page 11: Table 2: the kinetic formulations are "based on standard kinetic formulations in biogeochemical models (see text for details)". Can you include the relevant references in the table alongside each expression? I am not sure I can see them all in the text.

We have added the relevant references to Table 2.

-Figure 4: perhaps I am misreading this (?) but given that panel a has H2S at 100uM and panel b has Fe2 at 100um, then shouldn't the fractional values be the same for each plot when we look at 100uM on the x axis? E.g. Fe2(free) is 40% in panel a but only 20% in panel b under what appear to be the exact same conditions?

That is a good remark of the reviewer, there seemed to be indeed a conversion error when we plotted the visualMINTEQ results. We have rerun the simulations and corrected the corrected figure. This has not affected any primary conclusions.

-Page 18 line 15: might note recent work that proposes an important role for gypsum in Precambrian oxygenation e.g. Shields et al (2019) nature geosci.

Following a comment of reviewer 1, we have extended our discussion of the potential importance of gypsum and added a reference to the suggested paper at P19L21: '*We do not currently include precipitation of gypsum CaSO4 in our model description (Fig. 3b). Gypsum is an evaporite mineral that precipitates during regional and episodic events of supersaturation, and was likely a less important sulphur sink on a globally integrated basis during Precambrian time, or during any other period in which ocean SO42- was relatively low (Grotzinger et al., 1993, Crowe et al., 2014, Fakhraee et al., 2019). Indeed, there is still some debate as to the time-integrated impact of sulfate evaporites on the steady-state global sulfur cycle even during more recent periods of Earth's history (Halevy et al., 2012; Canfield, 2013). However, due to its episodic nature, gypsum could play an important role as a sulphate source during transient events (for example during events of enhanced weathering of a gypsum-rich source; Shields et al., 2019). Planned future developments to cGENIE will incorporate an explicit gypsum cycle, which would allow us to use the cGENIE.muffin model to investigate transient events of enhanced sulphate delivery to the ocean (see, e.g., Shields et al., 2019).*''

-Page 22: "Since the aim of this manuscript is to describe the newly developed Fe-S chemistry, and not ocean circulation in real or fake worlds, we will not further discuss the emerging patterns of ocean circulation." Yet they are discussed in the next section as they are fundamental for driving redox variability. There should be a brief summary of the circulation here.

We have added a brief summary of the important features of the emerging circulation: '*The sea surface temperature and ocean circulation generated by our configuration of cGENIE are shown in Fig. 3. Sea surface temperatures vary from ~30 °C at the equator to < 10 °C at the poles (Fig. 3c). The barotropic stream function shows a large degree of symmetry (Fig. 3d), whereas the overturning patterns are skewed to the southern hemisphere, with a strong anticlockwise circulation at around -60° N (Fig. 3e). The overturning stream function shows strong upwelling at the equator and deep-water mixing at the poles (Fig. 3e). The wind stress at the equator drives surface waters towards the west, which will lead to stronger upwelling on the west side of the continent. The redox patterns discussed below will reflect these two main features: (i) upwelling at the equator, in particular at the west side of the continent and (ii) deep-water mixing at the poles.*'

Figure 9 b, should say "S mass balance" as the title

Corrected

---

## Author Response (AR2)

Title: "Iron and sulphur cycling in the cGENIE.muffin Earth system model (v0.9.21)"
Tracking #: gmd-2020-312
Authors: van de Velde et al.

Topical editor:

Dear Authors,

Thank you for your revised manuscript and the responses to your referees.

I have examined both and am happy that the revisions address the issues raised during review. As such, I am very pleased to accept your manuscript for publication in GMD.

However, I did spot one extremely minor presentational issue in the manuscript. Panel b of Figure 3 ostensibly shows seafloor depth, but instead appears completely blacked-out. I understand that seafloor depth is homogeneous in the model, so the panel should appear "boring", but the colour map used for the panel does not obviously include black. Can you please amend this?

Going forwards, after this technical correction, the manuscript will be passed over to our production team. I am not formally involved at this stage, but if you feel that you need any assistance, please feel free to get back in contact.

Finally, thank you once again for selecting GMD for publishing your work.

With best regards,

Andrew Yool

We would like to thank the editor for his positive evaluation of our revision, and for this work as handling editor of our manuscript.
We changed the colour legend, so Fig. 3b is no longer 'just' black (even though that colour was included in the colour map, as the lowest value).

Kind regards
Sebastiaan van de Velde